# GET WHAT YOU WANT, NOT WHAT YOU DON'T: IMAGE CONTENT SUPPRESSION FOR TEXT-TO-IMAGE DIFFUSION MODELS

**Senmao Li**[1], **Joost van de Weijer**[2], **Taihang Hu**[1], **Fahad Shahbaz Khan**[3,4], **Qibin Hou**[1]
**Yaxing Wang**[1]*, **Jian Yang**[1]
[1]VCIP, CS, Nankai University, [2]Universitat Autònoma de Barcelona
[3]Mohamed bin Zayed University of AI, [4]Linkoping University
{senmaonk,hutaihang00}@gmail.com, joost@cvc.uab.es
fahad.khan@liu.se, {houqb,yaxing,csjyang}@nankai.edu.cn

## ABSTRACT

The success of recent text-to-image diffusion models is largely due to their capacity to be guided by a complex text prompt, which enables users to precisely describe the desired content. However, these models struggle to effectively suppress the generation of undesired content, which is explicitly requested to be omitted from the generated image in the prompt. In this paper, we analyze how to manipulate the text embeddings and remove unwanted content from them. We introduce two contributions, which we refer to as *soft-weighted regularization* and *inference-time text embedding optimization*. The first regularizes the text embedding matrix and effectively suppresses the undesired content. The second method aims to further suppress the unwanted content generation of the prompt, and encourages the generation of desired content. We evaluate our method quantitatively and qualitatively on extensive experiments, validating its effectiveness. Furthermore, our method is generalizability to both the pixel-space diffusion models (i.e. DeepFloyd-IF) and the latent-space diffusion models (i.e. Stable Diffusion).

## 1 INTRODUCTION

Text-based image generation aims to generate high-quality images based on a user prompt (Ramesh et al., 2022; Saharia et al., 2022; Rombach et al., 2021). This prompt is used by the user to communicate the desired content, which we call the *positive target*, and can potentially also include undesired content, which we define with the term *negative target*. Negative lexemes are ubiquitously prevalent and serve as pivotal components in human discourse. They are crucial for humans to precisely communicate the desired image content.

However, existing text-to-image models can encounter challenges in effectively suppressing the generation of the negative target. For example, when requesting an image using the prompt "a face without glasses", the diffusion models (i.e., SD) synthesize the subject without "glasses", as shown in Fig. 1 (the first column). However, when using the prompt "a man without glasses", both SD and DeepFloyd-IF models still generate the subject with "glasses" [1], as shown in Fig. 1 (the second and fifth columns). Fig. 1 (the last column) quantitatively show that SD has 0.819 *DetScore* for "glasses" using 1000 randomly generated images, indicating a very common failure cases in diffusion models. Also, when giving the prompt "a man", often the glasses are included, see Fig. 1 (the third and sixth columns). This is partially due to the fact that many of the collected man training images contain glasses, but often do not contain the *glasses* label (see in Appendix A Fig. 9).

Few works have addressed the aforementioned problem. The *negative prompt* technique[2] guides a diffusion model to exclude specific elements or features from the generated image. It, however, often

---

*The corresponding author.

[1]It also happens in both *Ideogram* and *Mijourney* models, see Fig. 27.

[2]The negative prompt technique is used in a text-to-image diffusion model to eliminate undesired content associated with a prompt inputted into the unconditional branch instead of using the null-text $\varnothing$.

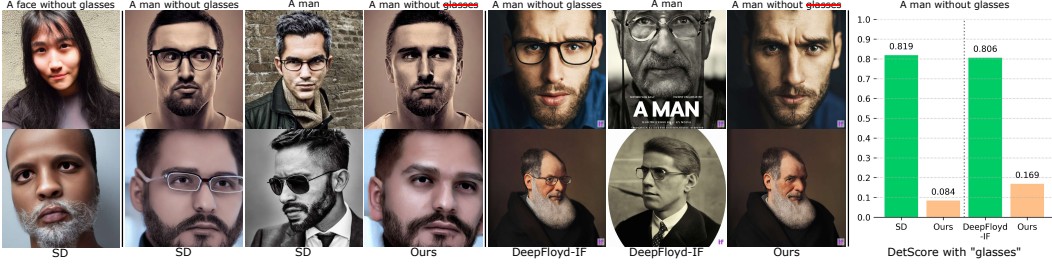

Figure 1: Failure cases of Stable Diffusion (SD) and DeepFloyd-IF. Given the prompt "A man without glasses", both SD and DeepFloyd-IF fail to suppress the generation of *negative target* glasses. Our method successfully removes the "glasses". (Right) we use DetScore (see Sec. 4) to detect the "glasses" from 1000 generated images. The DetScore of SD with prompt "A face without glasses" is 0.122. See Appendix E for additional examples.

leads to an unexpected impact on other aspects of the image, such as changes to its structure and style (see Fig. 6). Both P2P (Hertz et al., 2022) and SEGA (Brack et al., 2023) allow steering the diffusion process along several directions, such as weakening a target object from the generated image. We empirically observe these methods to lead to inferior performance (see Fig. 6 and Table 1 below). This is expected since they are not the tailored for this problem. Recent works (Gandikota et al., 2023; Kumari et al., 2023; Zhang et al., 2023) fine-tune the SD model to eliminate completely some targeted object information, resulting in catastrophic neglect (Kumari et al., 2022). A drawback of these methods is that the model is unable to generate this context in future text-prompts. Finally, Inst-Inpaint (Yildirim et al., 2023) requires paired images to train a model to erase unwanted pixels.

In this work, we propose an alternative approach for negative target content suppression. Our method does not require fine-tuning the image generator, or collecting paired images. It consists of two main steps. In the first step, we aim to remove this information from the text embeddings[3] which decide what particular visual content is generated. To suppress the negative target generation, we eliminate its information from the whole text embeddings. We construct a text embedding matrix, which consists of both the negative target and [EOT] embeddings. We then propose a *soft-weighted regularization* for this matrix, which explicitly suppresses the corresponding negative target information from the [EOT] embeddings. In the second step, to further improve results, we apply *inference-time text embedding optimization* which consists of optimizing the whole embeddings (processed in the first step) with respect to two losses. The first loss, called *negative target prompt suppression*, weakens the attention maps of the negative target further suppressing negative target generation. This may lead to the unexpected suppression of the positive target (see Appendix D. Fig. 13 (the third row)). To overcome this, we propose a *positive target prompt preservation* loss that strengthens the attention maps of the positive target. Finally, the combination of our proposed regularization of the text embedding matrix and the inference-time embedding optimization leads to improved negative target content removal during image generation.

In summary, our work makes the following **contributions**: (I) Our analysis shows that the [EOT] embeddings contain significant, redundant and duplicated semantic information of the whole input prompt (the whole embeddings). This needs to be taken into account when removing negative target information. Therefore, we propose soft-weighted regularization to eliminate the negative target information from the [EOT] embeddings. (II) To further suppress the negative target generation, and encourage the positive target content, we propose inference-time text embedding optimization. Ablation results confirm that this step significantly improves final results. (III) Through extensive experiments, we show the effectiveness of our method to correctly remove the negative target information without detrimental effects on the generation of the positive target content. Our code is available in `https://github.com/sen-mao/SuppressEOT`.

## 2 RELATED WORK

**Text-to-Image Generation.** Text-to-image synthesis aims to synthesize highly realistic images which are semantically consistent with the text descriptions. More recently, text-to-image models (Saharia et al., 2022; Ramesh et al., 2022; Rombach et al., 2021) have obtained amazing per-

---

[3]We use text embeddings to refer to the output from the CLIP text encoder.

formance in image generation. With powerful image generation capability, diffusion models allow users to provide a text prompt, and generate images of unprecedented quality. Furthermore, a series of recent works investigated knowledge transfer on diffusion models (Kawar et al., 2022; Ruiz et al., 2022; Valevski et al., 2022; Kumari et al., 2022) with one or few images. In this paper, we focus on the Stable Diffusion (SD) model without fientuning, and address the failure case that the generated subjects are not corresponding to the input text prompts.

**Diffusion-Based Image Generation.** Most recent works explore the ability to control or edit a generated image with extra conditional information, as well as text information. It contains label-to-image generation, layout-to-image generation and (reference) image-to-image generation. Specifically, label-to-image translation (Avrahami et al., 2022a;b; Nichol et al., 2021) aims to synthesize high-realistic images conditioning on semantic segmentation information, as well as text information. P2P (Hertz et al., 2022) proposes a mask-free editing method. Similar to label-to-image translation, both layout-to-image (Li et al., 2023b; Zhang & Agrawala, 2023) and (reference) image-to-image (Brooks et al., 2022; Parmar et al., 2023) generations aim to learn a mapping from the input image map to the output image. GLIGEN(Li et al., 2023b) boosts the controllability of the generated image by inserting bounding boxes with object categories. Some works investigate Diffusion-based inversion. (Dhariwal & Nichol, 2021) shows that a given real image can be reconstructed by DDIM (Song et al., 2020) sampling. Recent works investigate either the text embeddings of the conditional input (Gal et al., 2022; Li et al., 2023a; Wang et al., 2023), or the null-text optimization of the unconditional input (i.e., Null-Text Inversion (Mokady et al., 2022)).

**Diffusion-Based Semantic Erasion.** Current approaches (Gandikota et al., 2023; Kumari et al., 2023; Zhang et al., 2023) have noted the importance of erasure, including the erasure of copyright, artistic style, nudity, etc. ESD (Gandikota et al., 2023) utilizes negative guidance to lead the fine-tuning of a pre-trained model, aiming to achieve a model that erases specific styles or objects. (Kumari et al., 2023) fine-tunes the model using two prompts with and without erasure terms, such that the model distribution matches the erasure prompt. Inst-Inpaint (Yildirim et al., 2023) is a novel inpainting framework that trains a diffusion model to map source images to target images with the inclusion of conditional text prompts. However, these works fine-tune the SD model, resulting in catastrophic neglect for the unexpected suppression from input prompt. In this paper, we aim to remove unwanted subjects in output images without further training or fine-tuning the SD model.

## 3 METHOD

We aim to suppress the *negative target* generation in diffusion models. To achieve this goal, we focus on manipulating the text embeddings, which essentially control the subject generation. Naively eliminating a target text embedding fails to exclude the corresponding object from the output (Fig. 2a (the second and third columns)). We conduct a comprehensive analysis that shows this failure is caused by the appended [EOT] embeddings (see Sec. 3.2). Our method consists of two main steps. In the first step, we propose *soft-weighted regularization* to largely reduce the negative target text information from the [EOT] embeddings (Sec. 3.3). In the second step, we apply *inference-time text embedding optimization* which consists of optimizing the whole text embeddings (processed in the first step) with respect to two losses. The first loss, called the *negative target prompt suppression* loss, aims to weaken the attention map of the negative target to guide the update of the whole text embeddings, thus further suppressing the subject generation of the negative target. To prevent undesired side effects, namely the unexpected suppression from the positive target in the output (see Appendix D. Fig. 13 (the third row)), we propose the *positive target prompt preservation* loss. This strengthens the attention map of the positive target.The inference-time text embedding optimization is presented in Sec. 3.4. In Sec. 3.1, we provide a simple introduction to the SD model, although our method is not limited to a specific diffusion model.

### 3.1 PRELIMINARY: DIFFUSION MODEL

The SD firstly trains an encoder $E$ and a decoder $D$. The encoder maps the image $x$ into the latent representation $z_0 = E(x)$, and the decoder reverses the latent representation $z_0$ into the image $\hat{x} = D(z_0)$. SD trains a UNet-based denoiser network $\epsilon_\theta$ to predict noise $\epsilon$, following the objective:

$$\min_\theta E_{z_0, \epsilon \sim N(0,I), t \sim [1,T]} \|\epsilon - \epsilon_\theta(z_t, t, c)\|_2^2, \tag{1}$$

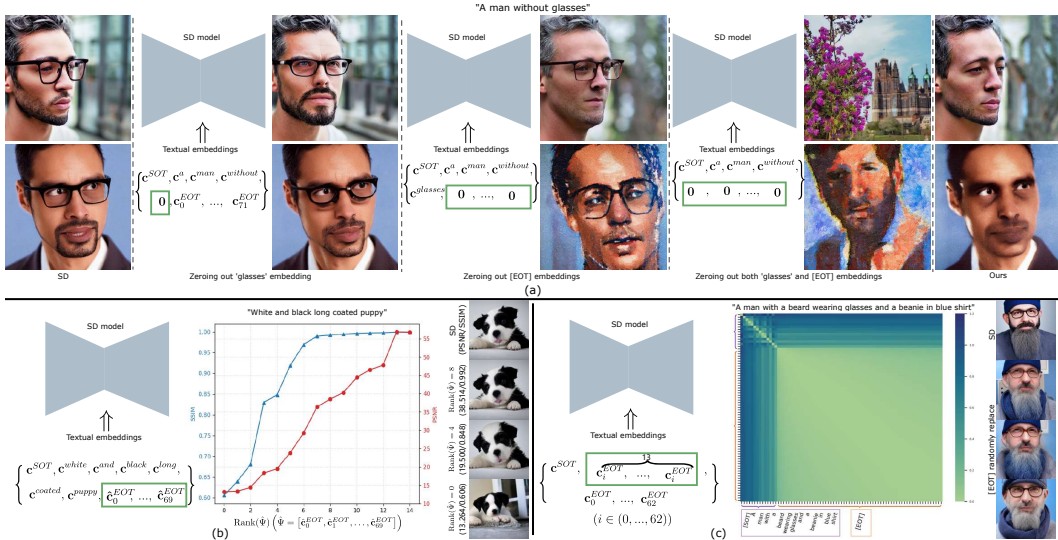

Figure 2: Analysis of [EOT] embeddings. (a) [EOT] embeddings contain significant information as can be seen when zeroed out. (b) when performing WNNM, we find that [EOT] embeddings have redundant semantic information. (c) distance matrix between all text embeddings. Note that each [EOT] embedding contains similar semantic information and they have near zero distance.

where the encoded text embeddings $c$ is extracted by a pre-trained CLIP text encoder $\Gamma$ with given a conditioning prompt $p$: $c = \Gamma(p)$, $z_t$ is a noise sample at timestamp $t \sim [1, T]$, and $T$ is the number of the timestep. The SD model introduces the cross-attention layer by incorporating the prompt. We could extract the internal cross-attention maps $A$, which are high-dimensional tensors that bind pixels and tokens extracted from the prompt text.

## 3.2 ANALYSIS OF [EOT] EMBEDDINGS

The text encoder $\Gamma$ maps input prompt $p$ into text embeddings $c = \Gamma(p) \in \mathbb{R}^{M \times N}$ (i.e., $M = 768$, $N = 77$ in the SD model). This works by prepending a *Start of Text* ([SOT]) symbol to the input prompt $p$ and appending $N - |p| - 1$ *End of Text* ([EOT]) padding symbols at the end, to obtain $N$ symbols in total. We define text embeddings $c = \{c^{SOT}, c_0^P, \cdots, c_{|p|-1}^P, c_0^{EOT}, \cdots, c_{N-|p|-2}^{EOT}\}$. Below, we explore several aspects of the [EOT] embeddings.

**What semantic information [EOT] embeddings contain?** We observe that [EOT] embeddings carry significant semantic information. For example, when requesting an image with the prompt "a man without glasses", SD synthesizes the subject including the negative target "glasses" (Fig. 2a (the first column)). When zeroing out the token embedding of "glasses" from the text embeddings $c$, SD fails to discard "glasses" (Fig. 2a (the second and third columns)). Similarly, zeroing out all [EOT] embeddings still generates the "glasses" subject (Fig. 2a (the fourth and fifth columns)). Finally, when zeroing out both "glasses" and the [EOT] token embeddings, we successfully remove "glasses" from the generated image (Fig. 2a (the sixth and seventh columns)). The results suggest that the [EOT] embeddings contain significant information about the input prompt. Note that naively zeroing them out often leads to unexpected changes (Fig. 2a (the seventh column)).

**How much information whole [EOT] embeddings contain?** We experimentally observe that [EOT] embeddings have the low-rank property [4], indicating they contain redundant semantic information. The weighted nuclear norm minimization (WNNM) (Gu et al., 2014) is an effective low-rank analysis method. We leverage WNNM to analyze the [EOT] embeddings. Specifically, we construct a [EOT] embeddings matrix $\Psi = [c_0^{EOT}, c_1^{EOT}, \cdots, c_{N-|p|-2}^{EOT}]$, and perform WNNM as follows $\mathcal{D}_w(\Psi) = U\mathcal{D}_w(\Sigma)V^T$, where $\Psi = U\Sigma V^T$ is the Single Value Decomposition (SVD) of $\Psi$, and $\mathcal{D}_w(\Sigma)$ is the generalized soft-thresholding operator with the weighted vector $w$, i.e.,

---

[4]This observation is based on the results of our statistical experiments on generated images, with a sample size of 100. The average PSNR is 49.300, SSIM is 0.992, and the average Rank($\hat{\Psi}$)=7.83.

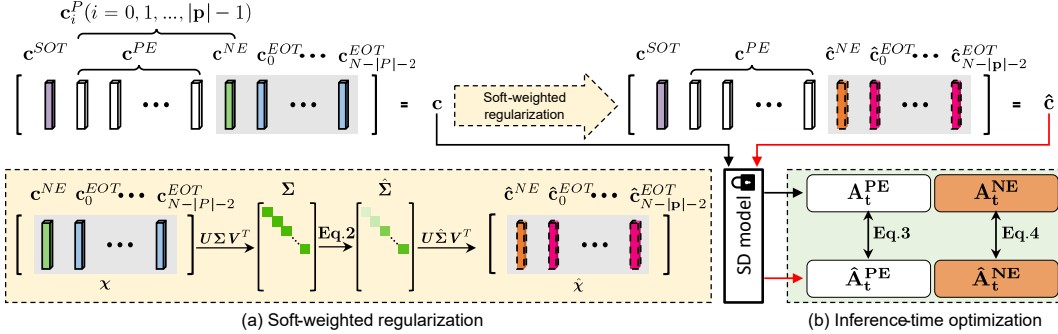

Figure 3: Overview of the proposed method. (a) We devise a negative target embedding matrix $\boldsymbol{\chi}$: $\boldsymbol{\chi} = [\boldsymbol{c}^{NE}, \boldsymbol{c}_0^{EOT}, \cdots, \boldsymbol{c}_{N-|\boldsymbol{p}|-2}^{EOT}]$. We perform SVD for the embedding matrix $\boldsymbol{\chi} = \boldsymbol{U\Sigma V}^T$. We introduce a soft-weight regularization for each largest eigenvalue.Then we recover the embedding matrix $\hat{\boldsymbol{\chi}} = \boldsymbol{U\hat{\Sigma} V}^T$. (b) We propose inference-time text embedding optimization (ITO). We align the attention maps of both $\boldsymbol{c}^{PE}$ and $\hat{\boldsymbol{c}}^{PE}$, and widen the ones of both $\boldsymbol{c}^{NE}$ and $\hat{\boldsymbol{c}}^{NE}$.

$\mathcal{D}_{\boldsymbol{w}}(\boldsymbol{\Sigma})_{ii} = \text{soft}(\boldsymbol{\Sigma}_{ii}, w_i) = \max(\boldsymbol{\Sigma}_{ii} - w_i, 0)$. The singular values $\boldsymbol{\sigma}_0 \geq \cdots \geq \boldsymbol{\sigma}_{N-|\boldsymbol{p}|-2}$ and the weights satisfy $0 \leq w_0 \leq \cdots \leq w_{N-|\boldsymbol{p}|-2}$.

To verify the low-rank property of [EOT] embeddings, WNNM mainly keeps the *top-K* largest singular values of $\boldsymbol{\Sigma}$, zero out the small singular values, and finally reconstruct $\hat{\Psi} = \left[\hat{\boldsymbol{c}}_0^{EOT}, \hat{\boldsymbol{c}}_1^{EOT}, \cdots, \hat{\boldsymbol{c}}_{N-|\boldsymbol{p}|-2}^{EOT}\right]$. We use Rank($\hat{\Psi}$) to represent the rank of $\hat{\Psi}$. We explore the impact of different Rank($\hat{\Psi}$) values on the generated image. For example, as shown in Fig. 2b, with the prompt "White and black long coated puppy" (here $|\boldsymbol{p}| = 6$), we use PSNR and SSIM metrics to evaluate the modified image against the SD model's output. Setting Rank($\hat{\Psi}$)=0, zeroing all [EOT] embeddings, the generated image preserves similar semantic information as when using all [EOT] embeddings. As Rank($\hat{\Psi}$) increases, the generated image gets closer to the SD model's output. Visually, the generated image looks similar to the one of the SD model with Rank($\hat{\Psi}$)=4. Achieving acceptable metric values (PSNR=40.288, SSIM=0.994) with Rank($\hat{\Psi}$)=9 in Fig. 2b (middle). The results indicate that the [EOT] embeddings have the low-rank property, and contain redundant semantic information.

**Semantic alignment for each [EOT] embedding**   There exist a total of $76 - |\boldsymbol{p}|$ [EOT] embeddings. However, we find that the various [EOT] embeddings are highly correlated, and they typically contain the semantic information of the input prompt. This phenomenon is demonstrated both qualitatively and quantitatively in Fig. 2c. For example, we input the prompt "A man with a beard wearing glasses and a beanie in blue shirt". We randomly select one [EOT] embedding to replace input text embeddings like Fig. 2c (left) [5]. The generated images have similar semantic information (Fig. 2c (right)). This conclusion also is demonstrated by the distance of each [EOT] embedding (Fig. 2c (middle)). Most [EOT] embeddings have small distance among themselves. In conclusion, we need to remove the negative target information from the $76 - |\boldsymbol{p}|$ [EOT] embeddings.

### 3.3   TEXT EMBEDDING-BASED SEMANTIC SUPPRESSION

Our goal is to suppress negative target information during image generation. Based on the aforementioned analysis, we must eliminate the negative target information from the [EOT] embeddings. To achieve this goal, we introduce two strategies, which we refer to as *soft-weighted regularization* and *inference-time text embedding optimization*. For the former, we devise a negative target embedding matrix, and propose a new method to regularize the negative target information. The inference-time text embedding optimization aims to further suppress the negative target generation of the target prompt, and encourages the generation of the positive target. We give an overview of the two strategies in Fig. 3.

---

[5] The selected [EOT] embedding is repeated $|\boldsymbol{p}|$ times

**Soft-weighted Regularization.** We propose to use Single Value Decomposition (SVD) to extract negative target information (e.g., glasses) from the text embeddings. Let $c = \{c^{SOT}, c_0^P, \cdots, c_{|\boldsymbol{p}|-1}^P, c_0^{EOT}, \cdots, c_{N-|\boldsymbol{p}|-2}^{EOT}\}$ be the text embeddings from CLIP text encoder. As shown in Fig. 3 (left), we split the embeddings $c_i^P (i = 0, 1, \cdots, |\boldsymbol{p}| - 1)$ into the negative target embedding set $c^{NE}$ and the positive target embedding set $c^{PE}$. Thus we have $c = \{c^{SOT}, c_0^P, \cdots, c_{|\boldsymbol{p}|-1}^P, c_0^{EOT}, \cdots, c_{N-|\boldsymbol{p}|-2}^{EOT}\} = \{c^{SOT}, c^{PE}, c^{NE}, c_0^{EOT}, \cdots, c_{N-|\boldsymbol{p}|-2}^{EOT}\}$. We construct a negative target embedding matrix $\chi$: $\chi = \left[c^{NE}, c_0^{EOT}, \cdots, c_{N-|\boldsymbol{p}|-2}^{EOT}\right]$. We perform SVD: $\chi = U\Sigma V^T$, where $\Sigma = diag(\sigma_0, \sigma_1, \cdots, \sigma_{n_0})$, the singular values $\boldsymbol{\sigma}_1 \geq \cdots \geq \boldsymbol{\sigma}_{n_0}$, $n_0 = \min(M, N - |\boldsymbol{p}| - 1)$. Intuitively, the negative target embedding matrix $\chi = \left[c^{NE}, c_0^{EOT}, \cdots, c_{N-|\boldsymbol{p}|-2}^{EOT}\right]$ mainly contains the expected suppressed information. After performing SVD, we assume that the main singular values are corresponding to the suppressed information (the negative target). Then, to suppress negative target information, we introduce soft-weighted regularization for each singular value [6]:

$$\hat{\sigma} = e^{-\sigma} * \sigma. \tag{2}$$

We then recover the embedding matrix $\hat{\chi} = U\hat{\Sigma}V^T$, here $\hat{\Sigma} = diag(\hat{\sigma_0}, \hat{\sigma_1}, \cdots, \hat{\sigma_{n_0}})$. Note that the recovered structure is $\hat{\chi} = \left[\hat{c}^{NE}, \hat{c}_0^{EOT}, \cdots, \hat{c}_{N-|\boldsymbol{p}|-2}^{EOT}\right]$, and $\hat{c} = \{c^{SOT}, c^{PE}, \hat{c}^{NE}, \hat{c}_0^{EOT}, \cdots, \hat{c}_{N-|\boldsymbol{p}|-2}^{EOT}\}$.

We consider a special case where we reset top-K or bottom-K singular values to 0. As shown on Fig. 4, we are able to remove the negative target prompt (e.g, glasses or beard) when setting the top-K (here, K= 2) singular values to 0. And the negative target prompt information is preserved when the bottom-K singular values are set to 0 (here, K=70). This supports our assumption that main singular values of $\chi$ are corresponding to the negative target information.

## 3.4 INFERENCE-TIME TEXT EMBEDDING OPTIMIZATION

As illustrated in Fig. 3 (right), for a specific timestep $t$, during the diffusion process $T \to 1$, we get the diffusion network output: $\epsilon_\theta(\tilde{z}_t, t, c)$, and the corresponding attention maps: $(A_t^{PE}, A_t^{NE})$, where $c = \{c^{SOT}, c^{PE}, c^{NE}, c_0^{EOT}, \cdots, c_{N-|\boldsymbol{p}|-2}^{EOT}\}$. The attention maps $A_t^{PE}$ are corresponding to $c^{PE}$, while $A_t^{NE}$ are corresponding to $c^{NE}$ which we aim to suppress. After *soft-weighted regularization*, we have the new text embeddings $\hat{c} = \{c^{SOT}, c^{PE}, \hat{c}^{NE}, \hat{c}_0^{EOT}, \cdots, \hat{c}_{N-|\boldsymbol{p}|-2}^{EOT}\}$. Similarly, we are able to get the attention maps: $(\hat{A}_t^{PE}, \hat{A}_t^{NE})$.

Here, we aim to further suppress the negative target generation, and encourage the positive target information. We propose two attention losses to regularize the attention maps, and modify the text embeddings $\hat{c}$ to guide the attention maps to focus on the particular region, which is corresponding to the positive target prompt. We introduce an *positive target prompt preservation* loss:

$$\mathcal{L}_{pl} = \left\| \hat{A}_t^{PE} - A_t^{PE} \right\|^2. \tag{3}$$

That is, the loss attempts to strengthen the attention maps of the positive target prompt at the timestep $t$. To further suppress generation for the negative target prompt, we propose the *negative target prompt suppression* loss:

$$\mathcal{L}_{nl} = - \left\| \hat{A}_t^{NE} - A_t^{NE} \right\|^2, \tag{4}$$

***Full objective.*** The full objective function of our model is:

$$\mathcal{L} = \lambda_{pl}\mathcal{L}_{pl} + \lambda_{nl}\mathcal{L}_{nl}, \tag{5}$$

where $\lambda_{pl}$=1 and $\lambda_{nl}$=0.5 are used to balance the effect of preservation and suppression. We use this loss to update the text embeddings $\hat{c}$.

For real image editing, we first utilize the text embeddings $c$ to apply Null-Text (Mokady et al., 2022) to invert a given real image into the latent representation. Then we use the proposed soft-weighted regularization to suppress negative target information from $c$ resulting in $\hat{c}$. Next, we

---

[6]The inspiration for Eq. 2 is explained in detail in the Appendix B .

Table 1: Comparison with baselines. The best results are in bold, and the second best results are underlined.

| Method | Real-image editing | | | | | | Generated-image editing | | | | | | |
| --- | --- | --- | --- | --- | --- | --- | --- | --- | --- | --- | --- | --- | --- |
| | Random negative target | | | Random negative target | | | Negative target: Car | | | Negative target: Tyler Edlin | | Negative target: Van Gogh | |
| | Clipscore↓ | IFID↑ | DetScore↓ | Clipscore↓ | IFID↑ | DetScore↓ | Clipscore↓ | IFID↑ | DetScore↓ | Clipscore↓ | IFID↑ | Clipscore↓ | IFID↑ |
| Real image or SD (Generated image) | 0.7986 | 0 | 0.3381 | 0.8225 | 0 | 0.4509 | 0.8654 | 0 | 0.6643 | 0.7414 | 0 | 0.8770 | 0 |
| Negative prompt | 0.7983 | **175.8** | 0.2402 | 0.7619 | 169.0 | 0.1408 | 0.8458 | 151.7 | 0.5130 | 0.7437 | 233.9 | 0.8039 | 242.1 |
| P2P (Hertz et al., 2022) | 0.7666 | 92.53 | 0.1758 | 0.8118 | 103.3 | 0.3391 | 0.8638 | 21.7 | 0.6343 | 0.7470 | 86.3 | 0.8849 | 139.7 |
| ESD (Gandikota et al., 2023) | - | - | - | - | - | - | 0.7986 | 165.7 | 0.2223 | **0.6954** | **256.5** | 0.7292 | 267.5 |
| Concept-ablation (Kumari et al., 2023) | - | - | - | - | - | - | 0.7642 | 179.3 | 0.0935 | 0.7411 | 211.4 | 0.8290 | 219.9 |
| Forget-Me-Not (Zhang et al., 2023) | - | - | - | - | - | - | 0.8701 | 158.7 | 0.5867 | 0.7495 | 227.9 | 0.8391 | 203.5 |
| Inst-Inpaint (Yildirim et al., 2023) | 0.7327 | 135.5 | 0.1125 | 0.7602 | 150.4 | 0.1744 | 0.8009 | 126.9 | 0.2361 | - | - | - | - |
| SEGA (Brack et al., 2023) | - | - | - | 0.7960 | 172.2 | 0.3005 | 0.8001 | 168.8 | 0.4767 | 0.7678 | 209.9 | 0.8730 | 175.0 |
| Ours | **0.6857** | 166.3 | **0.0384** | **0.6647** | **176.4** | 0.1321 | 0.7426 | **206.8** | **0.0419** | 0.7402 | 217.7 | **0.6448** | **307.5** |

apply inference-time text embedding optimization to update $\hat{c}_t$ during inference, resulting in the final edited image. Our full algorithm is presented in Algorithm 1. See Appendix C for more detail about negative target generation of SD model without the reference real image.

---

**Algorithm 1: Our algorithm**

**Input:** A text embeddings $c = \Gamma(p)$ and real image $\mathcal{I}$.
**Output:** Edited image $\hat{\mathcal{I}}$.

$\widetilde{z}_T = \text{Inversion}(E(\mathcal{I}), c)$ ; // e.g., Null-text
$\hat{c} \leftarrow \text{SWR}(c)$ (Eq. 2) ; // SWR
**for** $t = T, T-1 \ldots, 1$ **do**
$\quad \hat{c}_t = \hat{c}$;
$\quad$ // ITO
$\quad$ **for** $ite = 0, \ldots, 9$ **do**
$\quad\quad A_t^{PE}, A_t^{NE} \leftarrow \epsilon_\theta(\widetilde{z}_t, t, c)$;
$\quad\quad \hat{A}_t^{PE}, \hat{A}_t^{NE} \leftarrow \epsilon_\theta(\widetilde{z}_t, t, \hat{c}_t)$ ;
$\quad\quad \mathcal{L} \leftarrow \lambda_{pl}\mathcal{L}_{pl} + \lambda_{nl}\mathcal{L}_{nl}$(Eqs. 3-6);
$\quad\quad \hat{c}_t \leftarrow \hat{c}_t - \eta\nabla_{\hat{c}_t}\mathcal{L}$ ;
$\quad$ **end**
$\quad \widetilde{z}_{t-1}, _-, _- \leftarrow \epsilon_\theta(\widetilde{z}_t, t, \hat{c}_t)$
**end**
**Return** Edited image $\hat{\mathcal{I}} = D(\widetilde{z}_0)$

---

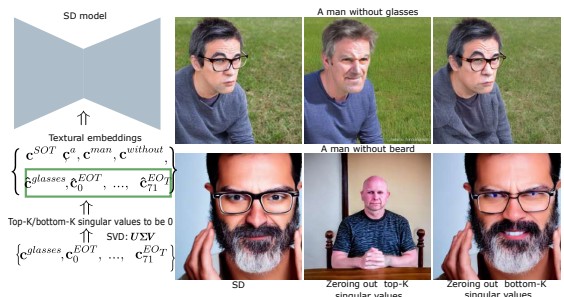

Figure 4: Effect of resetting *top-K* or *bottom-K* singular values to 0. Main singular values correspond to the target information that we expect to be suppressed.

## 4 EXPERIMENTS

**Baseline Implementations.** We compare with the following baselines: Negative prompt, ESD (Gandikota et al., 2023), Concept-ablation (Kumari et al., 2023), Forget-Me-Not (Zhang et al., 2023), Inst-Inpaint (Yildirim et al., 2023) and SEGA (Brack et al., 2023). We use P2P (Hertz et al., 2022) with **Attention Re-weighting**.

**Evaluation datasets.** We evaluate the proposed method from two perspectives: *generated-image editing* and *real-image editing*. In the former, we suppress the negative target generation from a generated image of the SD model with a text prompt, and the latter refers to editing a real-image input and a text input. Similar to recent editing-related works (Mokady et al., 2022; Gandikota et al., 2023; Patashnik et al., 2023), we use nearly 100 images for evaluation. For *generated-image* negative target surrpression, we randomly select 100 captions provided in the COCO's validation set (Chen et al., 2015) as prompts.The Tyler Edlin and Van Gogh related data (prompts and seeds) are obtained from the official code of ESD (Gandikota et al., 2023). For *real-image* negative target suppression, we randomly select 100 images and their corresponding prompts from the Unsplash [7] and COCO datasets.We also evaluate our approach on the GQA-Inpaint dataset, which contains 18,883 unique source-target-prompt pairs for testing. See Appendix. A for more details on experiments involving this dataset. We show the optimization details and more results in Appendix A, D and E, respectively.

**Metrics.** *Clipscore* (Hessel et al., 2021) is a metric that evaluates the quality of a pair of a negative prompt and an edited image. We also employ the widely used Fréchet Inception Distance (FID) (Heusel et al., 2017) for evaluation. To evaluate the suppression of the target prompt information after editing, we use inverted FID (*IFID*), which measures the similarity between two sets. In this metric, the larger the better. We also propose to use the DetScore metric, which is based on MMDetection (Chen et al., 2019) with GLIP (Li et al., 2022). We detect the negative target object in the edited image, successful editing should lead to a low DetScore (see Fig. 5 and Appendix A

---

[7] https://unsplash.com/

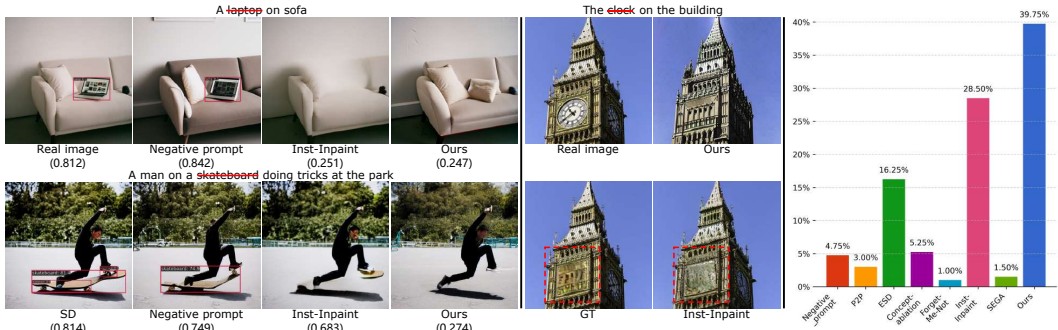

Figure 5: (Left) We detect the negative target from the edited images and and show the DetScore below. (Middle) Real image negative target suppression results. Inst-Inpaint fills the erased area with unrealistic pixels (the red dotted line frame). Our method exploits surrounding content information. (Right) User study.

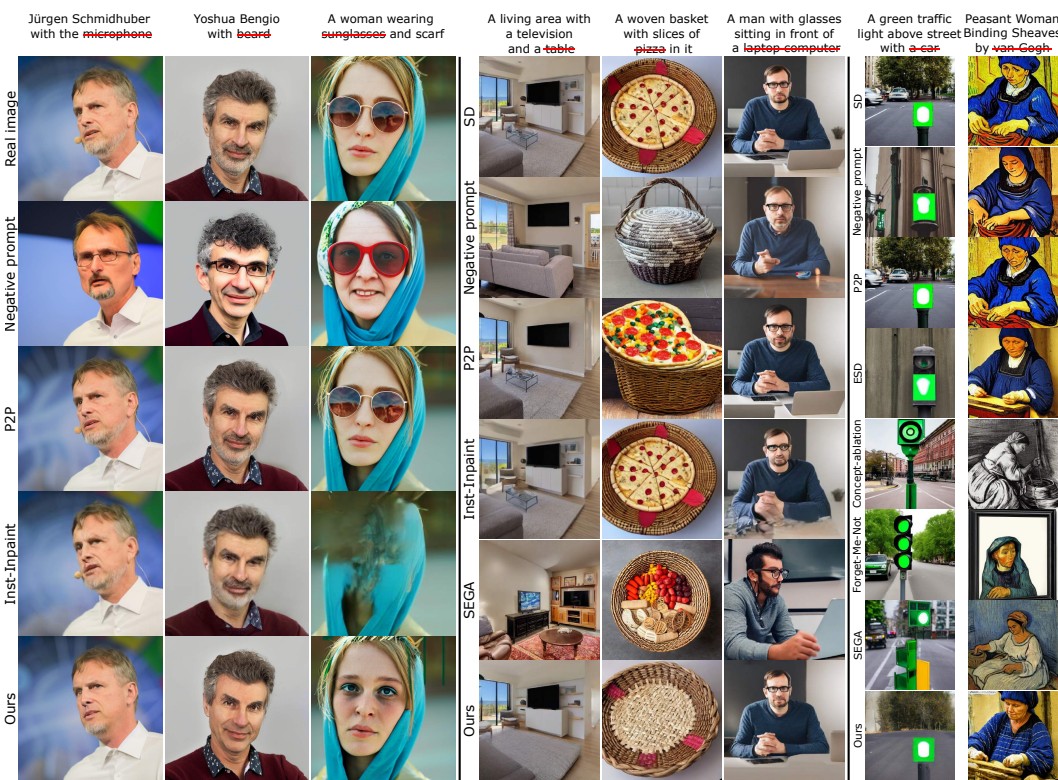

Figure 6: Real image (Left) and generated image (Middle and Right) negative target suppression results. (Middle) We are able to suppress the negative target, without further finetuning the SD model. (Right) Examples of negative target suppression.

for more detail). Following Inst-Inpaint (Yildirim et al., 2023), we use FID and CLIP Accuracy to evaluate the accuracy of the removal operation on the GQA-Inpaint dataset.

For real-image negative target suppression, as reported in Table 1 we achieve the best score in both Clipscore and DetScore (Table 1 (the second and fourth columns)), and a comparable result in IFID. Negative prompt has the best performance in IFID score. However, it often changes the structure and style of the image (Fig. 6 (left, the second row)). In contrast, our method achieves a better balance between preservation and suppression (Fig. 6 (left, the last row)). For generated image negative target suppression, we have the best performance for both a random and specific negative target, except for removing Tyler Edlin's style, for which ESD obtains the best scores. However, ESD requires to finetune the SD model, resulting in catastrophic neglect. Our advantage is further substantiated by visualized results (Fig. 6).

Figure 7: Additional applications. Our method can be applied to image restoration tasks, such as shadow, cracks, and rain removal. Also we can strengthen the object generation (6-9 column).

As shown in Fig. 5 (middle) and Table 2, we achieve superior suppression results and higher CLIP Accuracy scores on the GQA-Inpaint dataset. Inst-Inpaint achieves the best FID score (Table 2 (the third column)) primarily because its results (Fig. 5 (the second row, the sixth column)) closely resemble the ground truth (GT). However, the GT images contain unrealistic pixels. Our method yields more photo-realistic results. These results demonstrate that the proposed method is effective in suppressing the negative target. See Appendix. A for more experimental details.

**User study.** As shown in Fig. 5 (Right), we conduct a user study. We require users to select the figure in which the negative target is more accurately suppressed. We performed septuplets comparisons (forced choice) with 20 users (20 quadruplets/user). The results demonstrate that our method outperforms other methods. See Appendix. E for more details.

**Ablation analysis.** We conduct an ablation study for the proposed approach. We report the quantitative result in Table 3. Using soft-weighted regularization (SWR) alone cannot completely remove objects from the image. The results indicate that using both SWR and inference-time text embedding optimization leads to the best scores. The visualized results are presented in Appendix. D.

**Additional applications.** As shown in Fig. 7 (the first to the fifth columns), we perform experiments on a variety of image restoration tasks, including shadow removal, cracks removal and rain removal. Interestingly, our method can also be used to remove these undesired image artifacts. Instead of extracting the negative target embedding, we can also strengthen the added prompt and [EOT] embeddings. As shown in Fig. 7 (the sixth to the ninth columns), our method can be successfully adapted to strengthen image content, and obtain results that are similar to methods like GLIGEN (Li et al., 2023b) and Attend-and-Excite (Chefer et al., 2023) (See Appendix. F for a complete explanation and more results).

Table 2: Quantitative comparison on the GQA-Inpaint dataset for real image negative target suppression task.

| Methods | Paired data | FID ↓ | CLIP Acc ↑ | CLIP Acc (top5) ↑ |
|---|---|---|---|---|
| X-Decoder | ✓ | 6.86 | 69.9 | 46.5 |
| Inst-Inpaint | ✓ | **5.50** | 80.5 | 60.4 |
| Ours | ✗ | 13.87 | **92.8** | **83.3** |

Table 3: Ablation study. The effectiveness of both soft-weighted regularization and inference-time text embedding optimization.

| | Clipscore↓ | IFID↑ | DetScore↓ |
|---|---|---|---|
| SD | 0.8225 | 0 | 0.4509 |
| SWR | 0.7996 | 85.9 | 0.3668 |
| SWR+ $\mathcal{L}_{pl}$ | 0.8015 | 100.2 | 0.3331 |
| SWR + $\mathcal{L}_{pl} + \mathcal{L}_{nl}$ | **0.6647** | **176.4** | **0.1321** |

## 5 Conclusions and Limitations

We observe that diffusion models often fail to suppress the generation of negative target information in the input prompt. We explore the corresponding text embeddings and find that [EOT] embeddings contain significant, redundant and duplicated semantic information. To suppress the generation of negative target information, we provide two contributions: soft-weighted regularization and inference-time text embedding optimization. In the former, we suppress the negative target information from the text embedding matrix. The inference-time text embedding optimization encourages the postive target to be preserved, as well as further removing the negative target information. **Limitations:** Currently, the test-time optimization costs around half a minute making the proposed method unfit for applications that require fast results. But, we believe that a dedicated engineering effort can cut down this time significantly.

ACKNOWLEDGEMENTS

This work was supported by funding by projects TED2021-132513B-I00 and PID2022-143257NB-I00 funded by MCIN/AEI/ 10.13039/501100011033 and by the European Union NextGenerationEU/PRTR and FEDER. The project supported by Youth Foundation (62202243). Computation is supported by the Supercomputing Center of Nankai University.

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

## A  APPENDIX: IMPLEMENTATION DETAILS

**Configure.**   We suppress semantic information by optimizing whole text embeddings at inference time, and it takes as little as 35 seconds. No extra network parameters are required in our optimization process. We mainly use the Stable Diffusion v1.4 pre-trained model [8]. All of our experiments are conducted using a Quadro RTX 3090 GPU (24GB VRAM).

**Early stop.**   Recent works Hertz et al. (2022); Chefer et al. (2023) demonstrate that the spatial location of each subject is decided in the early step. Thus we validate our method on different steps in inference time. Fig 8 (left) shows that our method suffers from artifacts after 20 timesteps. In this paper, at inference time we apply the proposed method among $0 \rightarrow 20$ timesteps, and for the remaining timesteps, we perform the original image generation as done in the SD model.

**Inner iterations.**   Fig 8 (right) shows the generation at different iterations within each timestep. We observe that the output images undergo unexpected change after 10 iterations. We set the iteration number to 10.

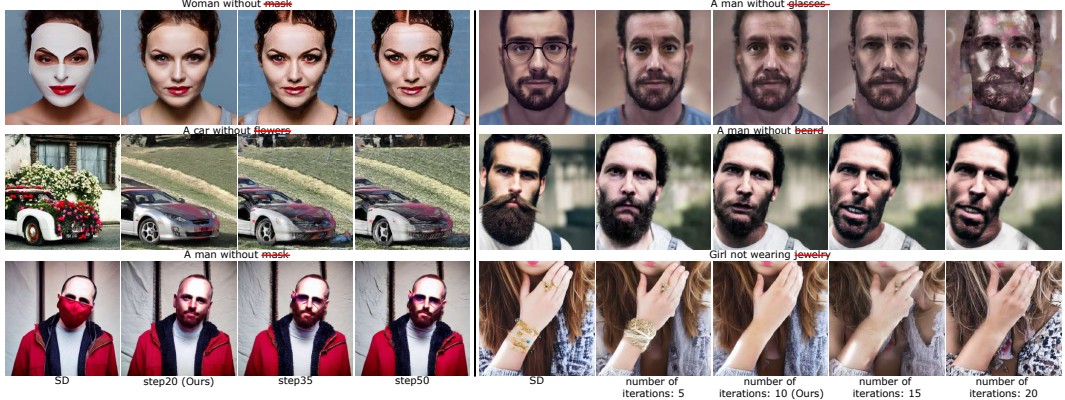

Figure 8: (Left) We stop optimizing at step 20 and keep the original model operating for the rest of the steps. (Right) The synthesized images with different iterations. We observe that we have better performance when setting iteration to 10.

**Inaccuracy label.**   Fig. 9 shows that the collected man training images contain glasses, but often do not contain the glasses label.

**IFID**   We use the official FID code to compute the similarity/distance between two distributions of image dataset, namely the edited images and the ground truth (GT) images. This measurement assesses the overall distribution rather than a single image. In the ideal case, our goal is to suppress only the content associated with the negative target in the image while leaving the content related to the positive target unaffected. We evaluate the effectiveness of suppression by comparing the FID values of the image dataset before and after suppression. A higher FID indicates a more successful suppression effect (referred to as IFID). However, we experimentally observed that many suppression methods (e.g., Negative prompt) can inadvertently impact the positive target while suppressing the negative target. Therefore, we will use IFID as the secondary metric, and Clipscore and DetScore as the primary metrics.

**DetScore**   We introduce a DetScore metric. It use MMDetection (Chen et al., 2019) with GLIP (Li et al., 2022) to detect the negative prompt object from the generated image and real image (e.g., the negative prompt object "laptop" in prompt "A laptop on sofa" in Fig. 5 (left)). We refer to the prediction score as *DetScore*. We set the prediction score threshold to 0.7, and our method achieves the best value in quantitative evaluation in both generated and real images (see Table 1 (the fourth, seventh and tenth columns)).

**Generated-image editing experiment details.**   The proposed method aims to focus attention on content suppression based on text embeddings, so we compare with various baselines on different

---

[8] https://huggingface.co/CompVis/stable-diffusion-v1-4

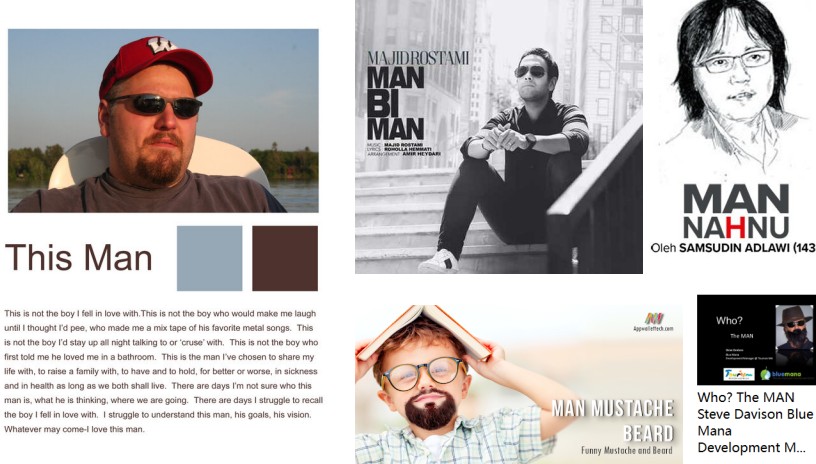

Figure 9: We find the collected man training images contain glasses, but often do not contain the glasses label.

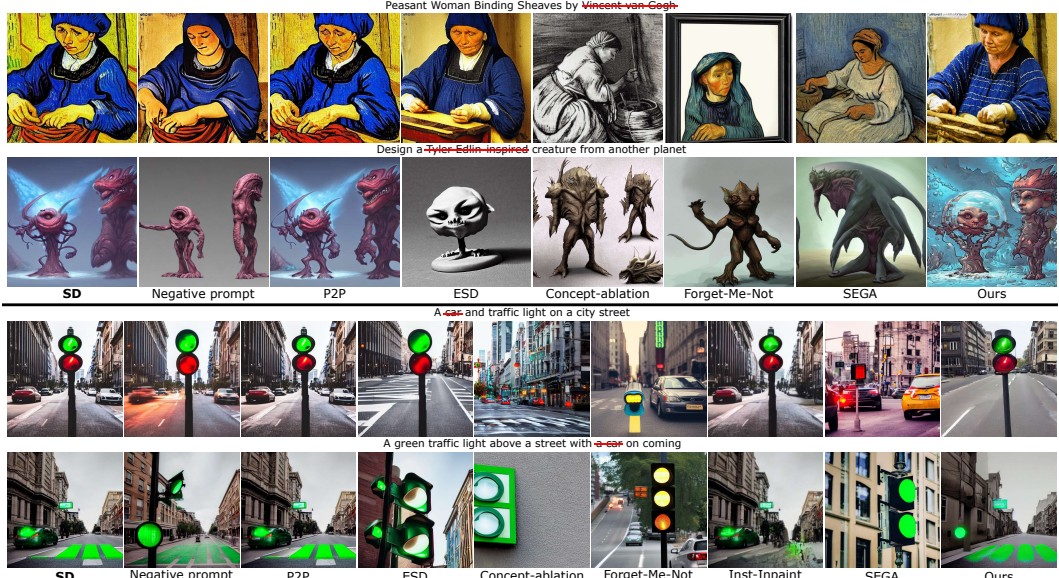

Figure 10: (Top) Comparisons with various baselines for generated images in the style of Van Gogh and Tyler Edlin. (Bottom) Comparisons with various baselines for generated car-related images.

types of generated images. **(1)** We compare our method with various baselines for generating images in the style of Van Gogh and Tyler Edlin (see Fig. 10 (Top) and Table 1 (the eleventh to the fourteenth columns)). The data related to Van Gogh and Tyler Edlin styles are sourced from the official code of ESD (Gandikota et al., 2023). This dataset comprises 50 prompts for Van Gogh style and 40 prompts for Tyler Edlin style. **(2)** To generate car-related images, we randomly select 50 car-related captions from COCO's validation set as prompts for input into SD. Additionally, we use multiple seeds for the same prompts. We chose to conduct experiments using car-related images for the specific reason that all baselines can effectively erase cars from the images, whereas the removal of other content is not universally suitable across all baselines. As shown in Table 1 (the eighth to the tenth columns), our method achieves the best values on the three evaluation metrics compared with all the baselines. Quantitative comparisons to various baselines are presented in Fig. 10 (Bottom). **(3)** For the other generated images used in the experiments (see Fig. 6 (the fourth to the sixth columns) and Table 1 (the fifth to the seventh columns)), we randomly select 100 captions provided in the COCO's validation set (Chen et al., 2015) as prompts, and input to the SD model.

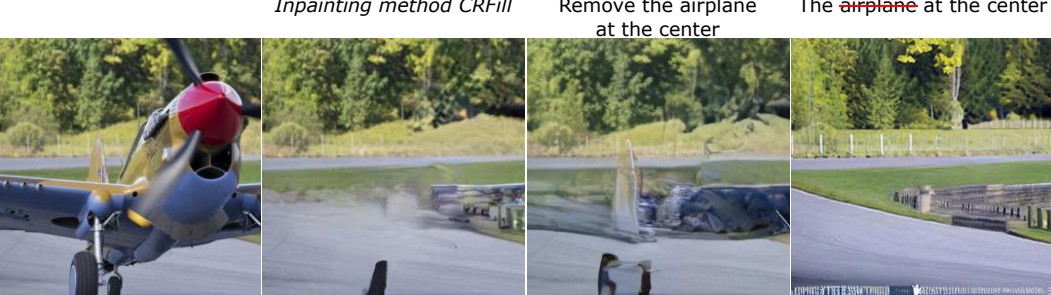

Figure 11: As an example, the instruction used in Inst-Inpaint is "Remove the airplane at the center", while our prompt is "The airplane at the center". GT is obtained using the image inpainting method CRFill (Zeng et al., 2021).

**GQA-Inpaint dataset experiment details.** Inst-Inpaint reports FID and CLIP Accuracy metrics for verification on the GQA-Inpaint dataset. FID compares the distributions of ground truth images (GT) and generated image distributions to assess the quality of images produced by a generative model. In the evaluation of Inst-Inpaint, the target image from the GQA-Inpaint dataset serves as the ground truth image when calculating FID. In Table 2 (the third column), Inst-Inpaint achieves the best FID score on the GQA-Inpaint dataset, primarily because the erasure results produced by Inst-Inpaint (Fig. 5 (the second row, the sixth column)) closely resemble the ground truth (GT) images (Fig. 5 (the second row, the fifth column)). Inst-Inpaint introduces CLIP Accuracy as a metric to assess the accuracy of the removal operation. For CLIP Accuracy, we use the official implementation of Inst-Inpaint. Inst-Inpaint use CLIP as a zero-shot classifier to predict the semantic labels of image regions based on bounding boxes. It compare the Top1 and Top5 predictions between the source image and inpainted image, considering a success when the source image class is not in the Top1 and Top5 predictions of the inpainted image. CLIP Accuracy is defined as the percentage of success. In Table 2 (the fourth and fifth columns), ours achieves the highest CLIP Accuracy scores for both Top1 and Top5 predictions on the GQA-Inpaint dataset. This result indicates the superior accuracy of our removal process.

Inst-Inpaint requires obtaining the target image corresponding to the source image as paired data for training. It extracts segmentation masks for each object from the source image and uses them to remove objects from the source image using the inpainting method CRFill (Zeng et al., 2021). The resulting target image is used as GT (e.g., Fig. 11 (the second column)).

There are 18883 pairs of test data in the GQA-Inpaint dataset, including source image, target image, and prompt. Inst-Inpaint attempts to remove objects from the source image based on the provided prompt as an instruction (e.g., "Remove the airplane at the center" in Fig. 11 (the third column)). We suppress the noun immediately following "remove" in the instruction (e.g., "airplane") and use the remaining part, deleting the word "remove" at the beginning of the instruction to form our input prompt (e.g., "The airplane at the center" in Fig. 11 (the fourth column)).

**Baseline Implementations.** For the comparisons in section 4, we use the official implementation of ESD (Gandikota et al., 2023) [9], Concept-ablation (Kumari et al., 2023) [10], Forget-Me-Not (Zhang et al., 2023) [11], Inst-Inpaint (Yildirim et al., 2023) [12] and SEGA (Brack et al., 2023) [13]. We use P2P (Hertz et al., 2022) [14] with **Attention Re-weighting** to weaken the extent of content in the resulting images.

**Failure cases.** Fig. 12 shows some failure cases.

---

[9] https://github.com/rohitgandikota/erasing
[10] https://github.com/nupurkmr9/concept-ablation
[11] https://github.com/SHI-Labs/Forget-Me-Not
[12] https://github.com/abyildirim/inst-inpaint
[13] https://github.com/ml-research/semantic-image-editing
[14] https://github.com/google/prompt-to-prompt

A ~~hamburger~~ and fries      A ~~giraffe~~ standing next to a bamboo building

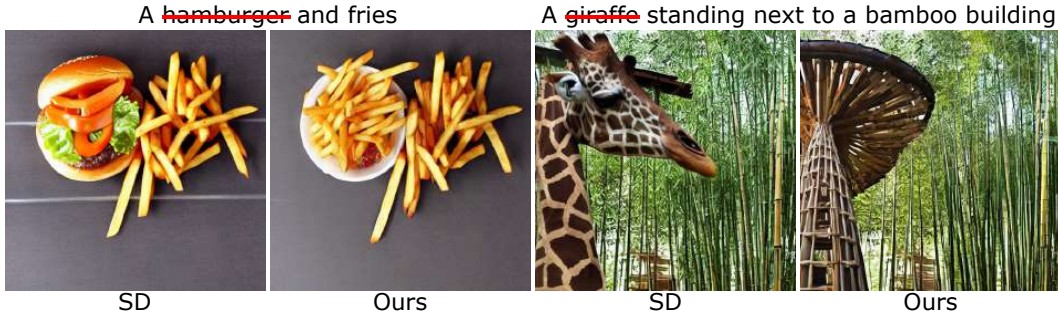

SD      Ours      SD      Ours

Figure 12: Failure cases.

## B  APPENDIX: EQ. 2 IN SOFT-WEIGHTED REGULARIZATION.

We take inspiration from WNNM, a method used for image denoising tasks, which demonstrates that singular values have a clear physical meaning and should be treated differently. WNNM considers that the noise in the image mainly resides in the *bottom-K* singular values. Each singular value $\sigma$ of the image patch can be updated using the formula $\sigma - \frac{\lambda}{(\sigma+\epsilon)}$ and set to 0 when the updated singular value becomes less than 0. The weight $\frac{\lambda}{(\sigma+\epsilon)}$ is introduced to ensure that components corresponding to smaller singular values undergo more shrinkage, where $\lambda$ is a positive constant used to scale the singular values, and $\epsilon$ is a small positive constant used to avoid division by zero. In this paper, based on our observation, the *top-K* singular values in the constructed negative target embedding matrix $\boldsymbol{\chi} = [\boldsymbol{c}^{NE}, \boldsymbol{c}_0^{EOT}, \cdots, \boldsymbol{c}_{N-|\boldsymbol{p}|-2}^{EOT}]$ mainly resides the content in the expected suppressed embedding $\boldsymbol{c}^{NE}$. Therefore, we utilize the formula $e^{-\sigma} * \sigma$ to ensure that the components corresponding to larger singular values undergo more shrinkage.

## C  APPENDIX: ALGORITHM DETAIL OF GENERATED IMAGE.

---

**Algorithm 2:** Our algorithm

**Require:** A text embeddings $\boldsymbol{c} = \Gamma(\boldsymbol{p})$ and noise vector $\widetilde{\boldsymbol{z}}_T$.
**Output:** Edited image $\hat{\mathcal{I}}$.

---

$\hat{\boldsymbol{c}} \leftarrow \text{SWR}(\boldsymbol{c})$ (Eq. 2) ;        // Soft-weighted Regularization
**for** $t = T, , T-1\ldots, 1$ **do**
     $\hat{\boldsymbol{c}}_{\boldsymbol{t}} = \hat{\boldsymbol{c}}$;
     // Inference-time text embedding optimization
     **for** $ite = 0, \ldots, Ite-1$ **do**
         $\text{-}, \boldsymbol{A}_{\boldsymbol{t}}^{\boldsymbol{PE}}, \boldsymbol{A}_{\boldsymbol{t}}^{\boldsymbol{NE}} \leftarrow \epsilon_\theta(\widetilde{\boldsymbol{z}}_t, t, \boldsymbol{c})$;
         $\text{-}, \hat{\boldsymbol{A}}_{\boldsymbol{t}}^{\boldsymbol{PE}}, \hat{\boldsymbol{A}}_{\boldsymbol{t}}^{\boldsymbol{NE}} \leftarrow \epsilon_\theta(\widetilde{\boldsymbol{z}}_t, t, \hat{\boldsymbol{c}}_{\boldsymbol{t}})$ (Eqs. 3-6);
         $\mathcal{L} \leftarrow min(\lambda_{pl}\mathcal{L}_{pl} + \lambda_{nl}\mathcal{L}_{nl})$;
         $\hat{\boldsymbol{c}}_{\boldsymbol{t}} \leftarrow \hat{\boldsymbol{c}}_{\boldsymbol{t}} - \eta\nabla_{\hat{\boldsymbol{c}}_{\boldsymbol{t}}}\mathcal{L}$ ;
     **end**
     $\widetilde{\boldsymbol{z}}_{t-1}, \text{-}, \text{-} \leftarrow \epsilon_\theta(\widetilde{\boldsymbol{z}}_t, t, \hat{\boldsymbol{c}}_{\boldsymbol{t}})$
**end**
$\hat{\mathcal{I}} = D(\widetilde{\boldsymbol{z}}_0)$
**Return** Edited image $\hat{\mathcal{I}}$

---

## D  APPENDIX: ABLATION ANALYSIS

**Verification alignment loss.** As shown in Fig. 13, $\mathcal{L}_{pl}$ can mainly hold regions that we do not want to suppress. In addition, we employ the SSIM metric to assess the influence of the $\mathcal{L}_{pl}$. Increasing

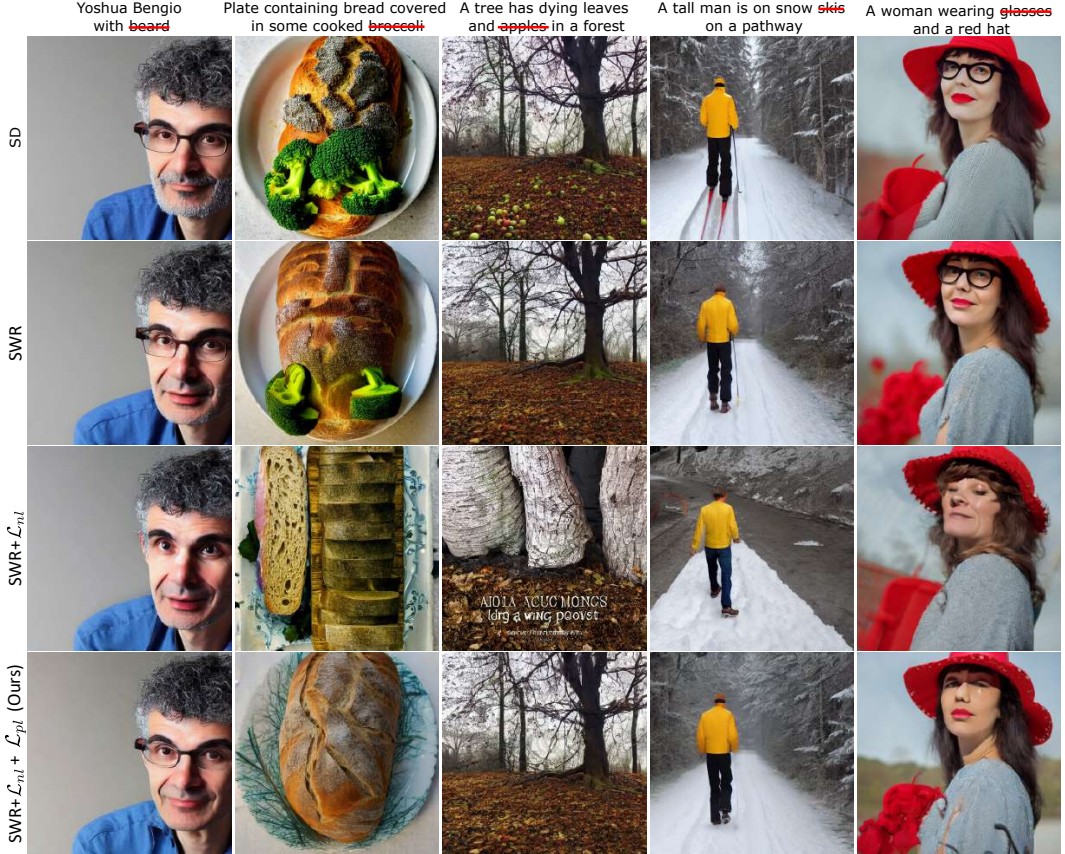

Figure 13: The regions that are not expected to be suppress are structurally altered without $\mathcal{L}_{pl}$ (third row). Our method removes the subject while mainly preserving the rest of the regions (fourth row).

$\mathcal{L}_{pl}$ raised SSIM from 0.407 (SWR+$\mathcal{L}_{nl}$) to 0.552 (SWR+$\mathcal{L}_{nl}$+$\mathcal{L}_{pl}$), indicating that $\mathcal{L}_{pl}$ can help preserve the rest of the regions. SWR+$\mathcal{L}_{nl}$, while capable of removing objects (**DetScore**=0.0692), tends to change the original image structure and style (**IFID**=242.3).

**Variant of soft-weighted regularization.** We also explore another way to regulate the target text embedding. We directly zero out the Top-K singular values of $\mathbf{\Sigma}$ ( here, $\mathbf{\Sigma} = diag(\sigma_0, \sigma_1, \cdots, \sigma_{n_0})$, $\mathbf{\chi} = \mathbf{U\Sigma V}^T$, $\mathbf{\chi} = [\mathbf{c}^{NE}, \mathbf{c}_0^{EOT}, \cdots, \mathbf{c}_{N-|\mathbf{p}|-2}^{EOT}]$ ), and reconstruct $\hat{\mathbf{\chi}}$, which is fed into SD model to generate image. Although directly zeroing out Top-K contributes to suppress the generation from the input prompt, it suffers from unexpected results (Fig. 14 (the third to the fifth columns)).

**Attention map to zero.** Recent work Hertz et al. (2022); Chefer et al. (2023); Parmar et al. (2023) explore the attention map to conduct varying tasks. In this paper, we also zero out the attention map which is corresponding to the target prompt, which is defined as *attn2zero*. As shown in Fig. 15, attn2zero method fails to suppress the target prompt in output images.

**Analysis of our method in long sentences** We use the object detection method to investigate the behavior of glasses when zeroing out both "glasses" and [EOT] embeddings in long sentences. We first randomly generate 1000 images using SD with the prompt $\mathbf{p}^{src}$ "A man without glasses" while generating a version that zeros out both "glasses" and [EOT] embeddings. We use MMDetection with GLIP and the prompt "glasses" to detect the probability of glasses being present in the generated images and obtain the prediction score for "glasses". The average prediction scores of MMDetection of the two versions above-mentioned on 1000 images are 0.819 and **0.084** (see Table 4 (third row, first and second column)), respectively, which proves that when using prompt $\mathbf{p}^{src}$, "A man without glasses", zeroing out the text embeddings of both "glasses" and [EOT] results in the disappearance of "glasses" in almost all generated images. It should be noted that the prediction

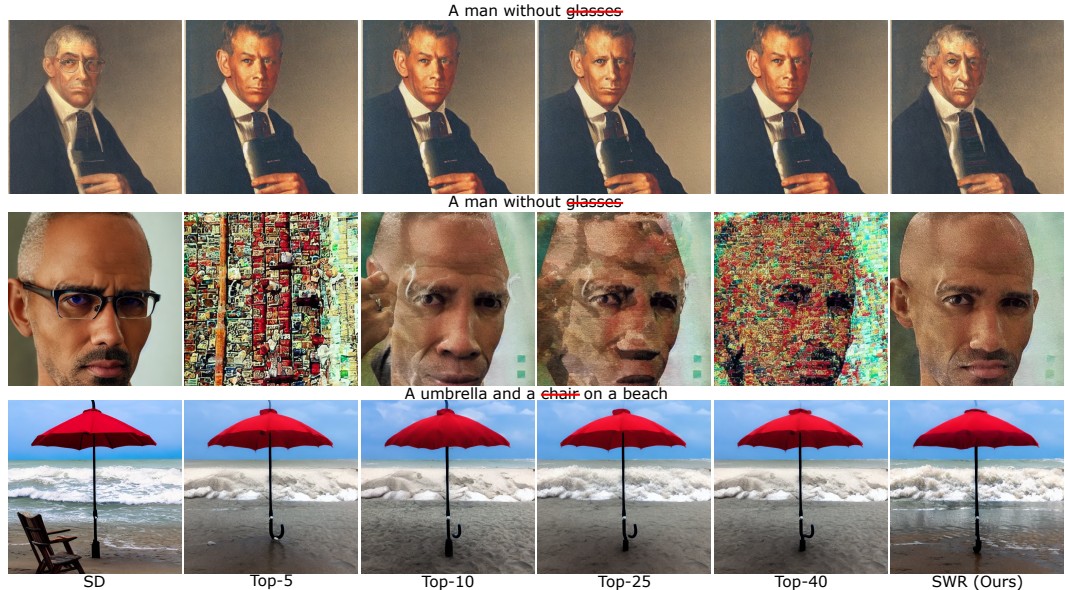

Figure 14: Variant of soft-weighted regularization. We zero out the Top-K singular values of $\Sigma$ ( $\Sigma = diag(\sigma_0, \sigma_1, \cdots, \sigma_{n_0})$ ). We experimentally observe that naively zeroing out the singular values suppresses the target prompt, but in some cases it leads to unwanted changes and expected results (the third to fifth columns).

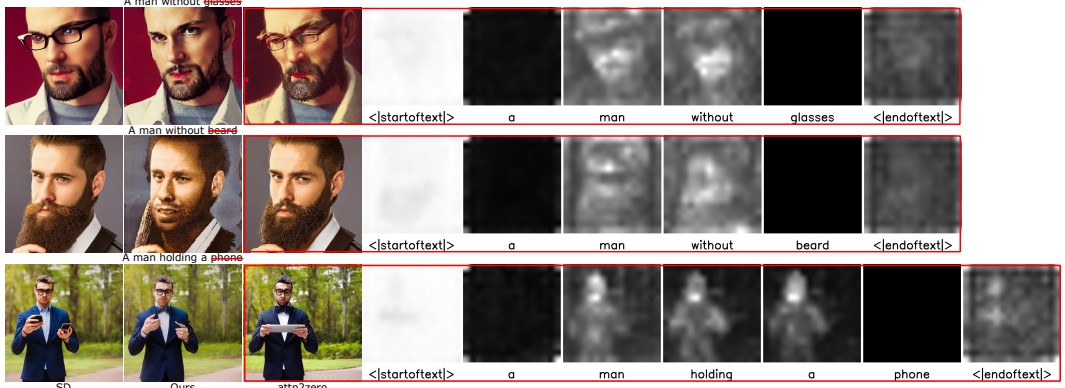

Figure 15: We set the attention map of the suppressed subject (e.g. glasses) to 0. We find it fail to remove this subject (third column). Ours successfully remove the subject (second column).

score of MMDetection does not indicate that $81.9\%$ of the 1000 images contain glasses. Instead, it represents the probability that the image is detected as containing glasses.

To investigate the behavior of glasses with long sentences, we use ChatGPT to generate description words of lengths 8, 16, and 32 after prompt $p^{src}$ to form new prompts denoted as $p^{src+8ws}$, $p^{src+16ws}$, and $p^{src+32ws}$, respectively. As shown in Table 4, when zeroing out both "glasses" and [EOT] embeddings, long sentences are harder to drop glasses than short sentences. This is due to the fact that other embeddings, except "glasses" and [EOT], contain more glasses information compared to short sentences. However, we observe that zeroing out both "glasses" and [EOT] embeddings works when most of the words in the prompt correspond to objects in the image, even when the sentence is long. (e.g. "A man with a beard wearing glasses and a hat in blue shirt") Therefore, our method requires a concise prompt that mainly describes the object, avoiding lengthy abstract descriptions.

Table 4: The average prediction score of MMDetection with GLIP using prompt "glasses".

| Method | SD | Zeroing out both "glasses" and [EOT] embeddings | | | |
|---|---|---|---|---|---|
| | $p^{src}$ | $p^{src}$ | $p^{src+8ws}$ | $p^{src+16ws}$ | $p^{src+32ws}$ |
| DetScore↓ | 0.819 | **0.084** | 0.393 | 0.455 | 0.427 |

**Different suppression levels for soft-weighted regularization.** We observe that the disappearance of the negative target (e.g., glasses in Fig. 2a (the sixth and seventh columns)) occurs when the negative target information diminishes to a certain level. We perform an analysis experiment to validate this conclusion. For example, we use $\gamma$ to control the suppression levels in soft-weighted regularization using $\hat{\sigma} = e^{-\gamma\sigma} * \sigma$ (in Eq. 2). When $\gamma = 0$, then $\hat{\sigma} = \sigma$, there is no change in singular values. When $\gamma = 1$, then $\hat{\sigma} = e^{-\sigma} * \sigma$, which equals to Eq. 2 that we used. When $\gamma \to \infty$, then $\hat{\sigma} = \lim_{\gamma \to \infty} e^{-\gamma\sigma} * \sigma = 0$, which equals to zero out both "glasses" and [EOT] embeddings in Fig. 2a (the sixth and seventh columns). As shown in Fig. 16, as $\gamma$ increases, the degree to which singular values are penalized gradually increases. When $\gamma$ increases to a certain level, at which the content of glasses in both the "glasses" and [EOT] embeddings decreases to a certain level, glasses will be erased.

A man without ~~glasses~~

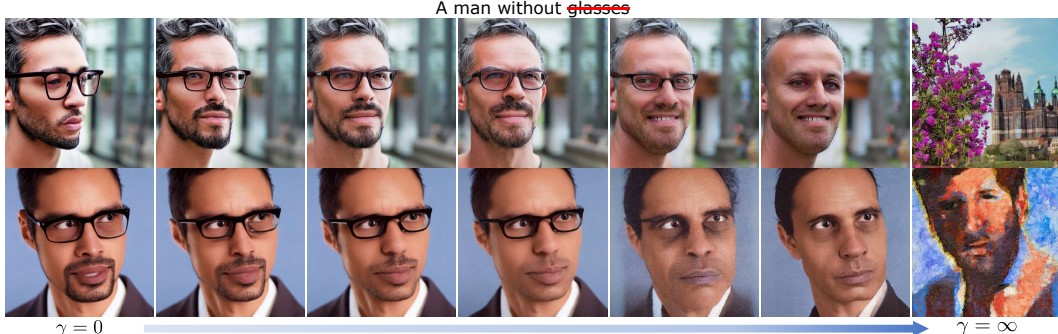

$\gamma = 0$          $\gamma = \infty$

Figure 16: Different suppression levels for soft-weighted regularization.

**Robustness to diverse input prompts.** As shown in Fig. 17, we showcase our robustness to diverse input prompts by effectively suppressing the content in an image using multiple prompts. It is important to emphasize that the suppressed content must be explicitly specified in the input prompt to enable our prompt-based content suppression.

A man with ~~a beard~~ wearing ~~glasses~~ and ~~a hat~~ in blue shirt     A man with ~~a beard~~     A man in ~~glasses~~     A man in ~~a beanie~~

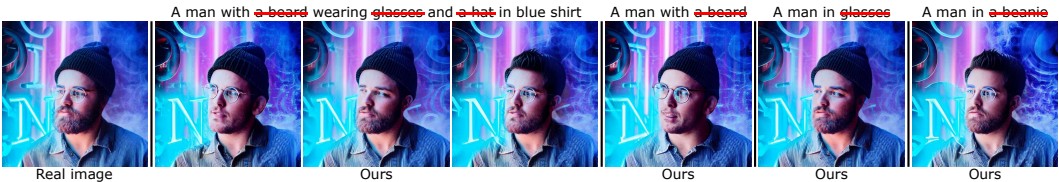

Real image      Ours      Ours      Ours      Ours

Figure 17: We can suppress the content using diverse input prompts.

**Evaluation the attenuation factor** We experimentally observed that employing an attenuation factor (e.g., 0.1) for the negative target embedding matrix would impact the positive target (see Fig. 20). Hence, using an attenuation factor leads to unexpected subject changes as well as changes to the target subject. This is due to the fact that the [EOT] embeddings contain significant information about the input prompt, including both the negative target and the positive target (see Sec. 3.2). Furthermore, the selection of factors needs to be carefully performed for each image to achieve satisfactory suppression results.

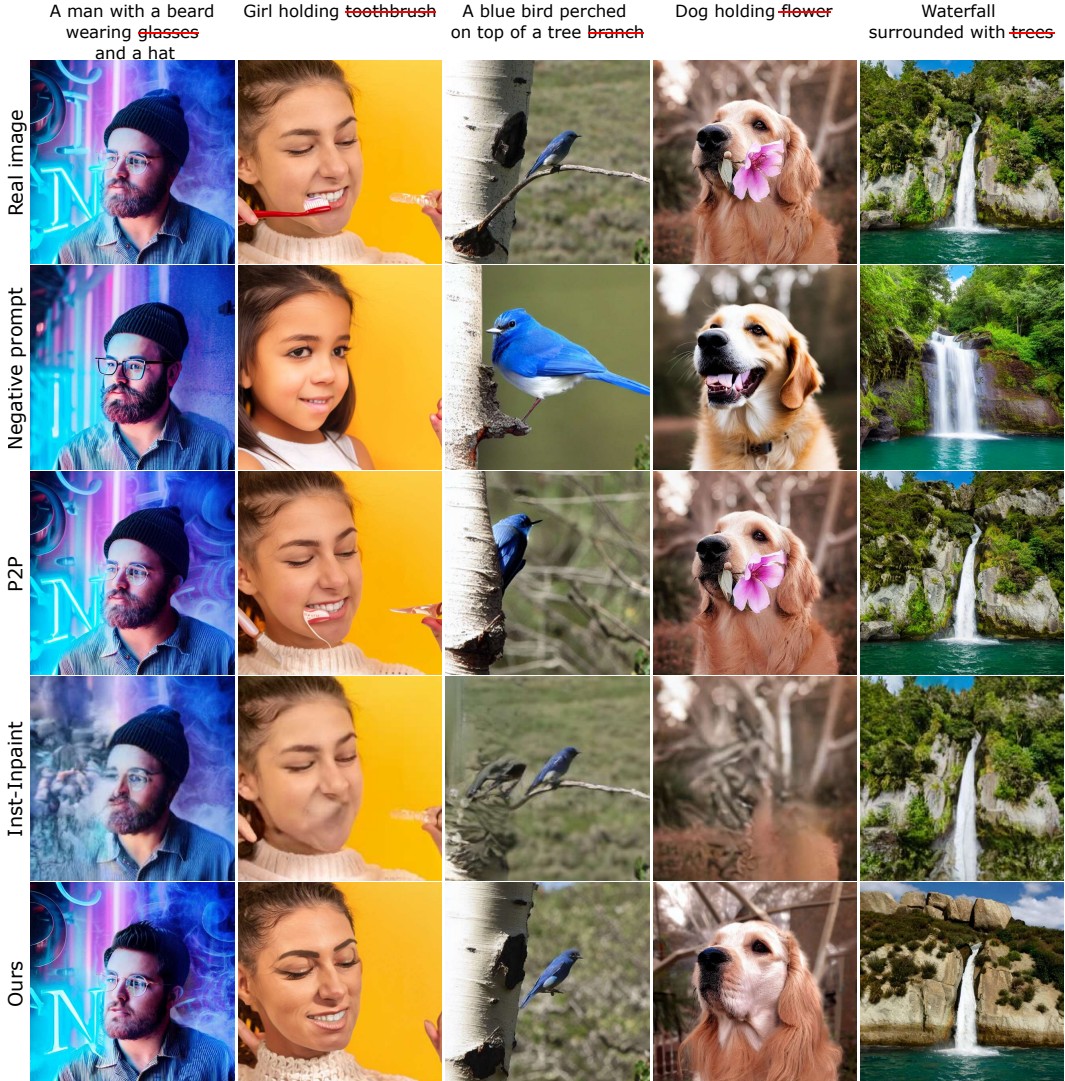

Figure 18: Additional reference-guided negative target generation results. Comparisons with various baselines for real image and the target prompt.

**[EOT] embedding in text prompts with various lengths.** We observe that the [EOT] embedding contains small yet useful semantic information, as demonstrated in Fig. 2c in our main paper. As shown in Fig. 2c, we randomly select one [EOT] embedding to replace input text embeddings. The generated images following this replacement have similar semantic information (see Fig. 2c). To further evaluate whether the [EOT] embedding contains useful semantic information in text prompts of various lengths, we replace the input text embeddings with not just one [EOT] embedding, but multiple. We use part of the [EOT] embedding when its length exceeds that of the input text embeddings (short sentence), and we copy multiple copies of the whole [EOT] embedding when its length is shorter than input text embeddings (long sentence).

In more detail, we first randomly chose 50 prompts from the prompt sets as mentioned in Sec. 4. These text prompts include various syntactical structures, such as "A living area with a television and a table", "A black and white cat relaxing inside a laptop" and "There is a homemade pizza on a cutting board". We add description words with lengths 8, 16, 32 and 56 following the initial text prompt $\mathbf{p}^{src}$ to obtain a long sentence, dubbed as $\mathbf{p}^{src+8ws}$, $\mathbf{p}^{src+16ws}$, $\mathbf{p}^{src+32ws}$, and $\mathbf{p}^{src+56ws}$, respectively. For instance, when $\mathbf{p}^{src}$ is "A living area with a television and a table", $\mathbf{p}^{src+8ws}$ would be extended to "A living area with a television and a table, highly detailed and precision with extreme detail description".

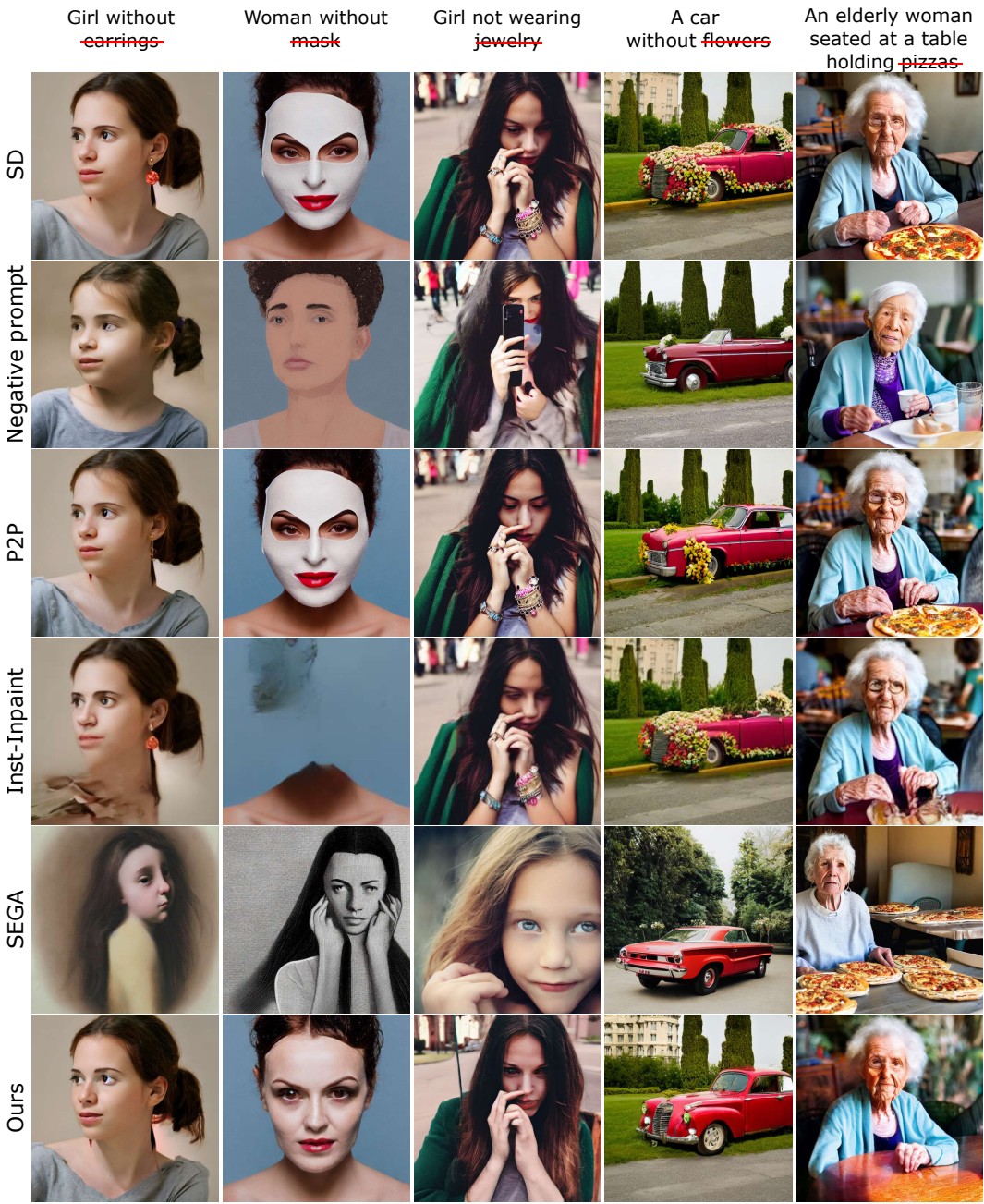

Figure 19: Additional latent-guided negative target generation results. Examples of our method and the baselines for generated image. We are able to suppress the target prompt, without further finetuning the SD model.

We use Clipscore to evaluate that the generated images match the given prompt. In this case, we test our model under various length prompts ($\mathbf{p}^{src}$, $\mathbf{p}^{src+8ws}$, $\mathbf{p}^{src+16ws}$, $\mathbf{p}^{src+32ws}$, and $\mathbf{p}^{src+56ws}$) (see Table 5 (the second and third rows)). As shown in Table 5, the generated images corresponding [EOT] embedding replacement prompts also contain similar semantic information compared to the initial prompt. The degeneration of the Clipscore is small (less than 0.11), indicating that the [EOT] embedding also contains semantic information. Fig. 21 shows some more qualitative results.

Related work also consider the [EOT] embedding for other tasks. For example, P2P manipulates the [EOT] attention injection when conducing image-to-image translation. P2P (Hertz et al., 2022)

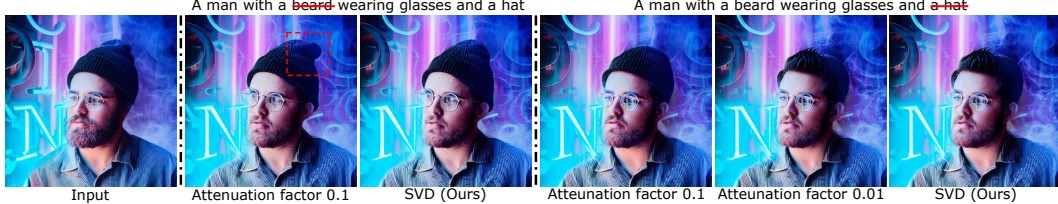

Figure 20: SWR with an attenuation factor and SVD. Note how the usage of an attenuation factor leads to undesired changes in the hat of the man (the second column).

Table 5: Comparison results with original tokens and their replacement version. We evaluate it with Clipscore.

| Mehod | $\mathbf{p}^{src}$ | $\mathbf{p}^{src+8ws}$ | $\mathbf{p}^{src+16ws}$ | $\mathbf{p}^{src+32ws}$ | $\mathbf{p}^{src+56ws}$ |
|---|---|---|---|---|---|
| SD | 0.8208 | 0.8173 | 0.8162 | 0.8102 | 0.8058 |
| SD w/ replacement | 0.7674 | 0.7505 | 0.7479 | 0.7264 | 0.7035 |

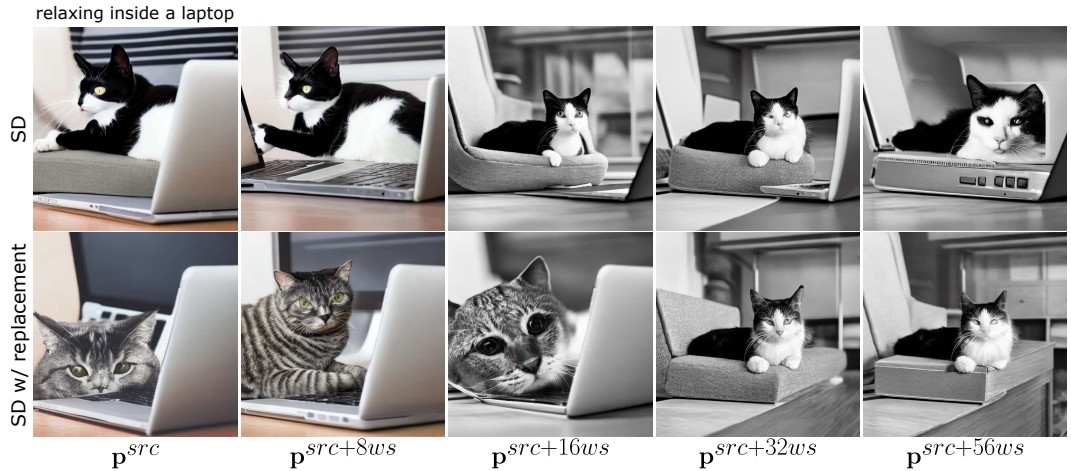

Figure 21: Both SD and its w/ replacement results.

swaps whole embeddings attention, including both the input text embeddings and [EOT] embedding attentions.

**Taking a simple mean of the [EOT] embedding** We extract the semantic component by taking a simple Mean of the Padding Embedding ([EOT] embedding), referred as **MPE**. We evaluate the propsed method (i.e., SVD) and **MPE**. We suppress "glasses" subject from 1000 randomly generated images with the prompt "A man without glasses". Then we use MMDetection detect the probability of glasses in the generated images. Final, we report the prediction score (DetScore).

As reported in Table 6 (the third and fourth columns), we have **0.1065** MMDetection score, while MPE is 0.6266. This finding suggests that simply averaging the [EOT] embedding often fails to extract the main semantic component. Furthermore, we further zero the 'glasses' token embedding as well as MPE, it still struggles to extract 'glasses' information (0.4892 MMDetection). Fig. 22 qualitatively shows more results.

**Inference-time optimization with value regulation.** We propose inference-time embedding optimization to further suppress the negative target generation and encourage the positive target content, following soft-weighted regularization. This optimization method involves updating the whole text embedding, which is then transferred to both the key and value components in the cross-attention layer. Therefore, our method implicitly changes the value component in the cross-attention layer.

Table 6: Comparison between ours and MPE. We report Clipscore.

| Mehod | SD | Ours | MPE | MPE + zeroing embedding |
|---|---|---|---|---|
| DetScore ↓ | 0.8052 | **0.1065** | 0.6266 | 0.4892 |

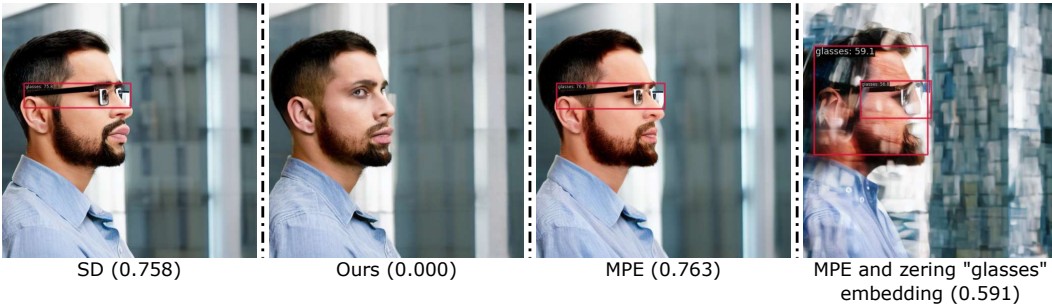


SD (0.758)     Ours (0.000)     MPE (0.763)     MPE and zering "glasses" embedding (0.591)


Figure 22: The visualization and DetScore when using a mean of the [EOT] embedding.

Furthermore, similar to the proposed two attention losses, we attempt to use two value losses to regulate the value component in the cross-attention layer:

$$\mathcal{L}_{vl} = \lambda_{pl}\mathcal{L}_{pl} + \lambda_{nl}\mathcal{L}_{nl},$$
$$\mathcal{L}_{pl} = \left\| \hat{V}_t^{PE} - V_t^{PE} \right\|^2,$$
$$\mathcal{L}_{nl} = - \left\| \hat{V}_t^{NE} - V_t^{NE} \right\|^2, \tag{6}$$

where hyper-parameters $\lambda_{pl}$ and $\lambda_{nl}$ are used to balance the effects of preservation and suppression of the value. When utilizing this value loss, we find that it is hard to generate high-quality images images (Fig. 23 (the third and sixth columns)). This result indicates that directly optimizing the value embedding does not work. The potential reason is that it also influences positive target, since each token embedding contains other token embedding information after CliptextEncoder.


A man without ~~glasses~~             A man with a beard wearing ~~glasses~~ and a beanie in blue shirt


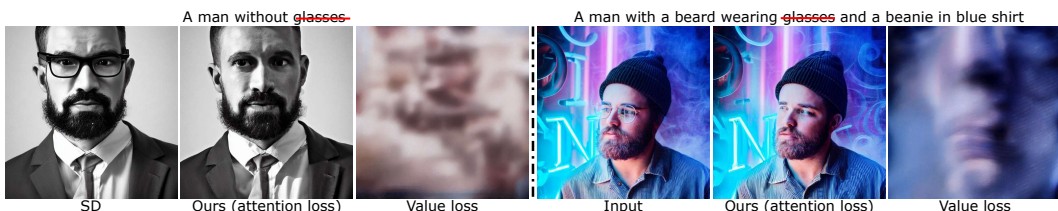


SD    Ours (attention loss)    Value loss    Input    Ours (attention loss)    Value loss


Figure 23: The results for generated image (left) and real image (right) of attention loss and value loss in the inference-time embedding optimization.

# E    APPENDIX: ADDITIONAL RESULTS

**User study.** The study participants were volunteers from our college. The questionnaire consisted of 20 questions, each presenting the original image generated by SD, as well as the results of various baselines and our method. Users are tasked with selecting an image in which the target subject (i.e., a car) is more accurately suppressed compared to the original image. Each question in the questionnaire presents eight options, including baselines (Negative prompt, P2P, ESD, Concept-ablation, Forget-Me-Not, Inst-Inpaint and SEGA) and our method, from which users were instructed to choose one. A total of 20 users participated, resulting in a combined total of 400 samples (20 questions × 1 option × 20 users), with 159 samples (39.75%) favoring our method (see Fig. 5

(Right)). In the results of the user study, the values for Ours, Negative Prompt, P2P, ESD, Concept-Ablation, Forget-Me-Not, Inst-Inpaint, and SEGA are 0.3975, 0.0475, 0.03, 0.1625, 0.0525, 0.01, 0.285, and 0.015, respectively.

**Additional results in our approach method.** Fig. 18 shows additional *real-image editing* results, and Fig. 19 shows additional *generated-image editing* results. It should be noted that the generated images, as shown in Fig. 19 (the first to fourth columns. i.e., "Girl without earring", "Woman without mask", "Girl not wearing jewelry" and "A car without flowers".), are not used for quantitative evaluation metrics in Table 1 (the fifth to the seventh columns), as the occasional failure of Clipscore (Hessel et al., 2021) to recognize negative words.

**Real image results in mask-based methods.** Mask-based removal methods work well for isolated objects. However, they tend to fail for objects that are closely related to their surroundings. Compared to mask-based methods, our prompt-based method can automatically complete regions of removed content based on surrounding content and works equally well when removed content is closely related to surrounding content. For example, in Fig. 24, the prompt "A man with a beard wearing glasses and a hat in blue shirt" and the corresponding input image show that "beard", "glasses", and "hat" are closely related to the man (Left). Our method can successfully remove "beard", "glasses", and "hat", and fill in the removed area based on the context of the "man" (Meddle), while the mask-based removal method appears very aggressive (Right).

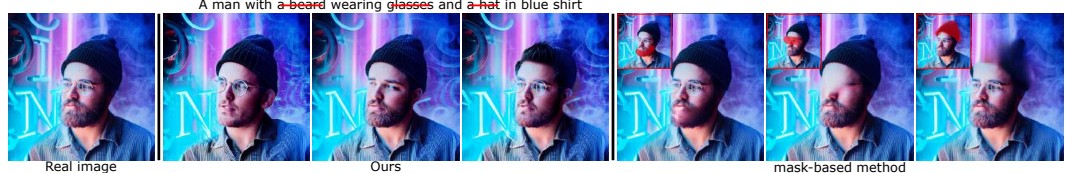

Figure 24: Ours can successfully remove "beard", "glasses", and "hat" and fill in the removed area based on the context of the "man" (Meddle), while the mask-based method (e.g., PlaygroundAI) fails (Right). The method reliant on masking necessitates the provision of user-specified masks that define the erased areas during the inference process.

**Real image results in various inversion methods.** Our method can combine various real image inversion techniques, including Null-text, Textual inversion mentioned in the Appendix (Textual inversion with a pivot.) of Null-text, StyleDiffusion (Li et al., 2023a), NPI (Miyake et al., 2023) and ProxNPI (Han et al., 2023) (see Fig. 25).

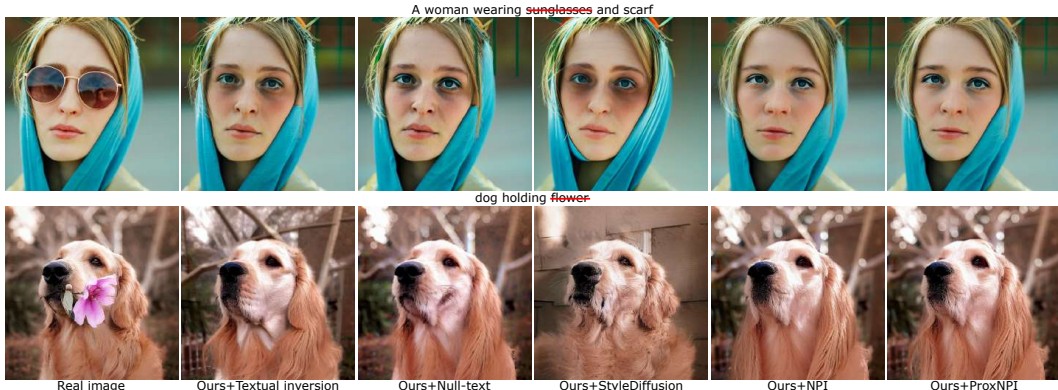

Figure 25: Our method can combine various real image inversion techniques.

**Implementation on DeepFloyd-IF diffusion model.** We use Deepfloyd-IF based on the T5 transformer to extract text embeddings using the prompt "a man without glasses" for generation. The generated output still includes the subject with "glasses" (see Fig. 26 (Up)), although the T5 text encoder used in Deepfloyd-IF has a larger number of parameters compared to the CLIP text encoder used in SD (T5: **4762.31M** vs. CLIP: 123.06M). Our method also works very well on DeepFloyd-IF diffusion model (see Fig. 26 (Bottom)).

A man without glasses

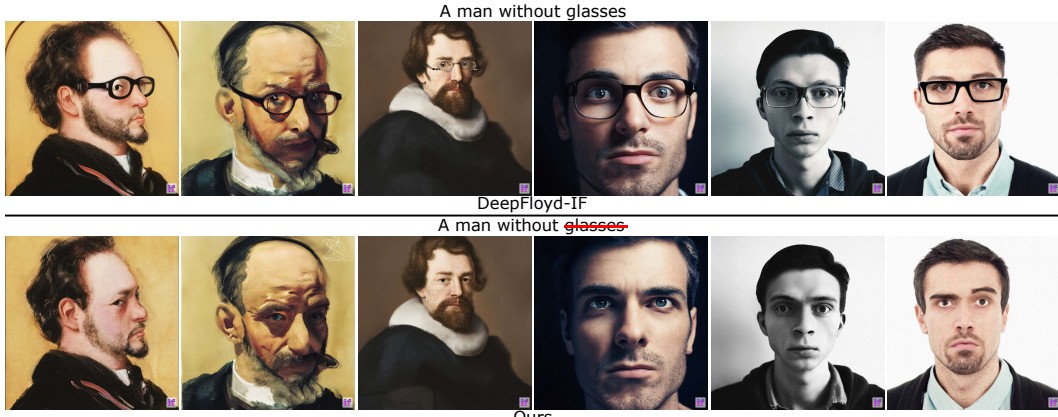

Figure 26: (Up) Results from DeepFloyd-IF still generate the man wearing "glasses". (Bottom) Implementation of our method on DeepFloyd-IF.

**"A man without glasses" results on other diffusion models.** When we use other diffusion models for image generation with the prompt "A man without glasses" as input, the generated images still show the presence of "glasses" (see Fig. 27 (Top)). Our method can also be implemented in other versions of the StableDiffusion model, including StableDiffusion 1.5 [15] and StableDiffusion 2.1 [16] (see Fig. 27 (Bottom)).

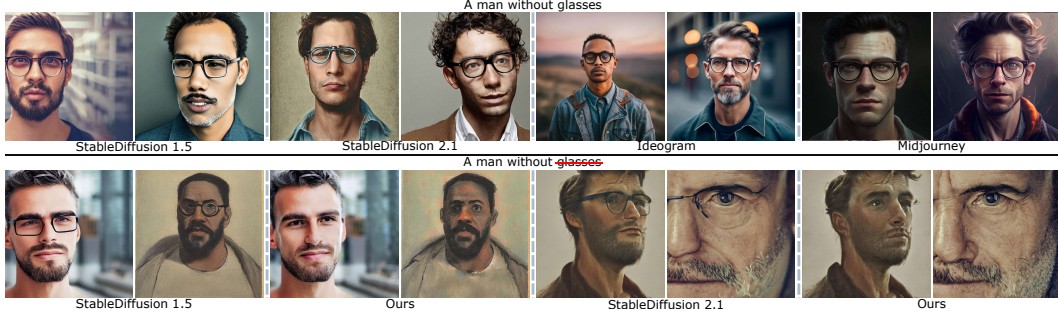

Figure 27: (Top) Results from StableDiffusion 1.5, StableDiffusion 2.1, Ideogram and Midjourney still generate the man wearing "glasses". (Bottom) Our method's implementation on StableDiffusion 1.5 and StableDiffusion 2.1.

## F APPENDIX: ADDITIONAL APPLICATIONS

**Additional cracks removal and rain removal results.** As shown in Fig. 28, we present additional results for both cracks removal and rain removal. (Up) Additional results for cracks removal. (Middle) We demonstrate additional results for the synthetic rainy image. (Down) Additionally, we also demonstrate additional results for the real-world rainy image.

**Attend-and-Excite similar results (Generating subjects for generated image).** Attend-and-Excitet (Chefer et al., 2023) find that the SD model sometimes encounters failure in generating one or more subjects from the input prompt (see Fig. 29 (the first, third, and fifth columns)). They refine the cross-attention map to attend to subject tokens and excite activations. The Eq. 2 used in soft-weighted regularization utilizes the weight $e^{-\sigma}$ to ensure that the components corresponding to larger singular values undergo more shrinkage, as we assume that the main singular values are corresponding to the suppressed information. We make a simple modification to the weight $e^{-\sigma}$ in

---

[15]https://huggingface.co/runwayml/stable-diffusion-v1-5
[16]https://huggingface.co/stabilityai/stable-diffusion-2-1

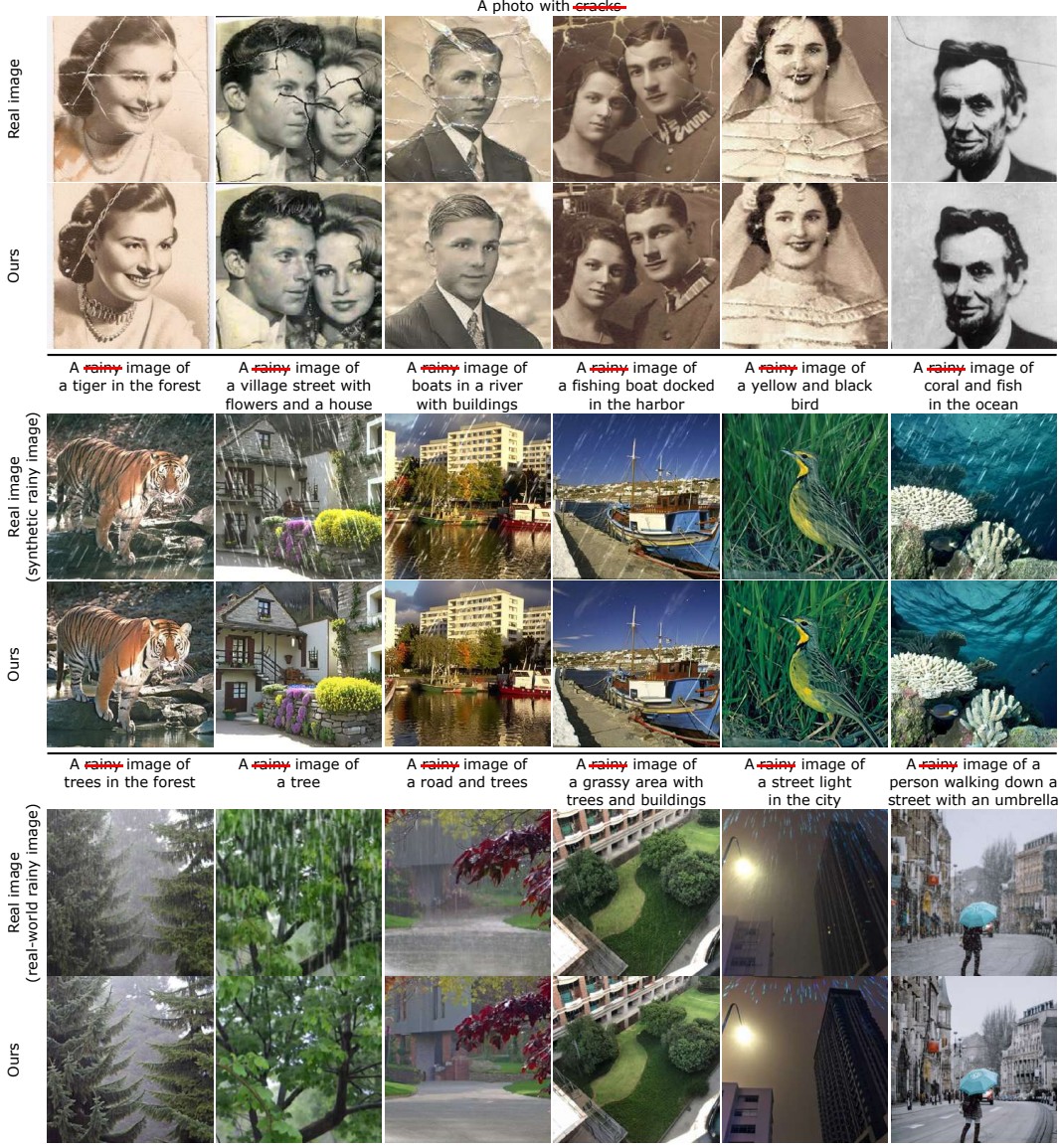

Figure 28: (Top) Cracks removal results. (Middle) Rain removal for synthetic rainy image. (Bottom) Rain removal for real-world rainy image.

Eq. 2 by using $\beta \cdot e^{\alpha\sigma}$ to ensure that the components corresponding to larger singular values undergo more strengthen (i.e., $\hat{\sigma} = \beta \cdot e^{\alpha\sigma} * \sigma$), where $\beta = 1.2$ and $\alpha = 0.001$. This straightforward modification, merely involving the update of text embeddings, addresses situations where the SD model encounters failures in generating subjects (see Fig. 29 (the second, fourth, and sixth columns)).

**GLIGEN similar results (Adding subjects for real image).** GLIGEN (Li et al., 2023b) can enable real image grounded inpainting, allowing users to integrate reference images into the real image. We can achieve results similar to real-image grounded inpainting using only the prompt (see Fig. 30 (the second, third, fifth, and sixth columns)). In detail, we add the prompt (blue underline) of the desired subject to the prompt describing the real image and then adopt the same strategy as in the previous subsection (**Attend-and-Excite similar results**).

**Replacing subject in the real image with another.** Subject replacement is a common task in various image editing methods Meng et al. (2021); Parmar et al. (2023); Mokady et al. (2022); Li et al. (2023a); Tumanyan et al. (2023). We can edit an image by replacing subject with another using only the prompt (see Fig. 31 (the second, fourth, and sixth columns)). We replace the text of the

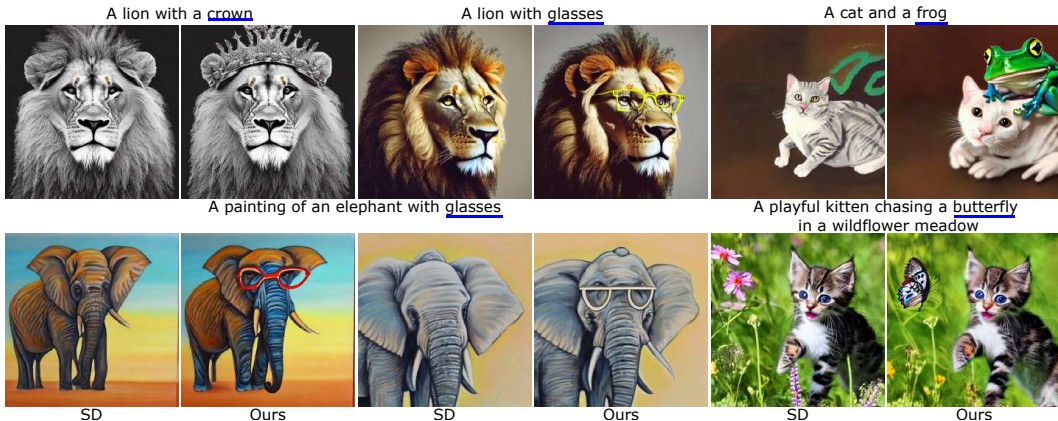

Figure 29: Attend-and-Excite similar results (Generating subjects for generated image).

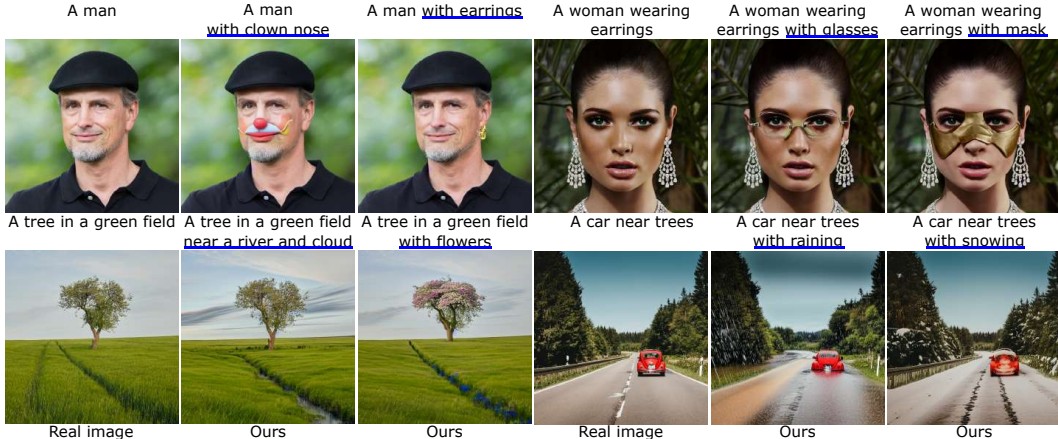

Figure 30: GLIGEN similar results (Adding subjects for real image).

edited subject in the source prompt with the desired one to create the target prompt. Subsequently, we translate the real image using the target prompt into latent code. We then apply the same strategy as in the previous subsection (**Attend-and-Excite similar results**) to obtain the edited image. For example, we can replace the "toothbrush" in the "Girl holding toothbrush" image with the "pen". The DetScore with "toothbrush" of the source image is 0.790, and the Clipscore with "pen" of the target image is 0.728. We find that the brace is also replaced with the pen, as the cross attention map for "toothbrush" couples the toothbrush and the brace.

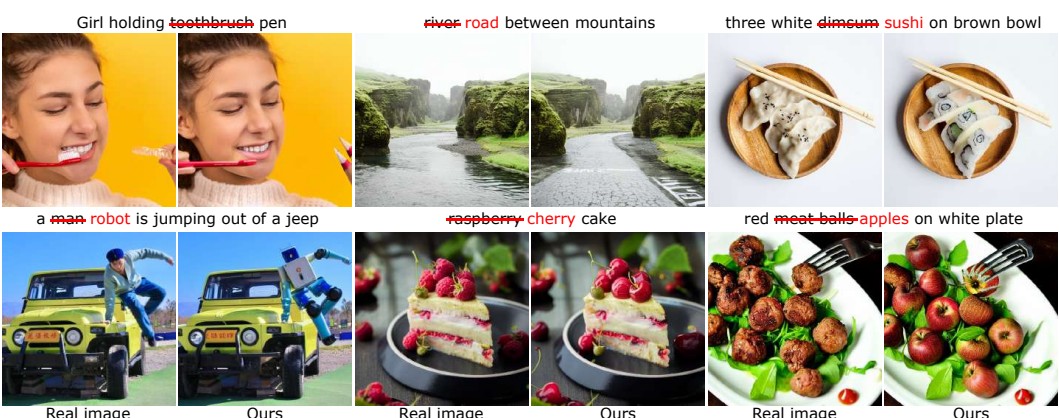

Figure 31: Replacing subject in the real image with another.

