# OpenReview forum: "Get What You Want, Not What You Don't: Image Content Suppression for Text-to-Image Diffusion Models"
_ICLR.cc/2024/Conference — ICLR 2024 poster_

### Official Review · Reviewer_B9jb · 2023-10-31

**Soundness:** 2 fair
**Presentation:** 3 good
**Contribution:** 2 fair
**Rating:** 6
**Confidence:** 4

**Summary:**

This paper proposes a method to eliminate side effects from padding promt tokens. 1. First it reveals that some side effects are hidden in the padding tokens, then it proposes an SVD based method to remove the side effects from the padding tokens. 2. It proposes a negative target prompt suppression loss to weaken the attention of the negative target in the prompt, 2. It proposes a positive target prompt preservation loss to avoid mistakenly suppressing positive targets in the prompt.

**Strengths:**

1. The analysis of the hidden semantics in padding tokens is interesting.
2. The design of the positive preservation and negative suppression losses are intuitive to understand.

**Weaknesses:**

*** Update after author responses ***
The authors addressed most of my concerns. Although I still feel the conclusions are a bit counter-intuitive, I'd like to raise my rating to 6, before seeing more cross validation from the community.

====== original comments ======

1. I'm not totally convinced that semantics in padding tokens have so much impact. My own empirical experience is that the padding tokens usually have very small attention scores (=> close to 0 attention probabilities) compared to meaningful tokens, and thus their semantics, if any, add little to the image features. Though, due to the large number of padding tokens, it might accumulate to somewhat significant impact, esp. when the prompt is short. This needs more systematic experiments to confirm, e.g. a diagram of the padding token impact w.r.t. the prompt length, where the prompts are randomly drawn from a pool.
2. All the padding tokens are derived from the same input word embedding, and only differ in the positional encoding added to the word embedding. If you want to extract the main semantic component, why not take a simple mean of the padding embeddings? Why using SVD is advantageous?

**Questions:**

1. We know cross attention consists of the attention map and the value recombination (output = v*attn). Even if the attention map values are largely suppressed, the undesired semantics may still slip into the image features through the value recombination. Have the authors tried to address this issue?
2. CLIPscores in Table 1 are a bit confusing. Are they the similarity between the images and negative prompts?

---

> ### Author Response · Authors · 2023-11-22
> **Response to Reviewer B9jb (1/3)**
>
> Thanks for your constructive comments.
>
> **1. More systematic experiments for the prompt with various length**
>
> **1.1. Replacing meaningful tokens with padding tokens.**
>
> We agree that the padding token contains less semantic information compared to meaningful tokens.
> However, we observe that the padding token contains small yet useful semantic information, as demonstrated in Fig.2c in our main paper.
> As shown in Fig.2c, we randomly select one padding token to replace all meaningful tokens. The generated images following this replacement have similar semantic information (see Fig.2c).
> To further evaluate whether the padding token contains useful semantic information in text prompts of various lengths, we replace the meaningful tokens with not just one padding token, but multiple.
> We use part of the padding token when its length exceeds that of the meaningful token (short sentence), and we copy multiple copies of the whole padding token when its length is shorter than meaningful token (long sentence).
>
> In more detail, we first randomly chose 50 prompts from the prompt sets as mentioned in Sec.4 in our main paper. These text prompts include various syntactical structures, such as "A living area with a television and a table", "A black and white cat relaxing inside a laptop" and "There is a homemade pizza on a cutting board".
> We add description words with lengths 8, 16, 32 and 56 following the initial text prompt $\mathbf{p}^{src}$ to obtain a long sentence, dubbed as $\mathbf{p}^{src+8ws}$, $\mathbf{p}^{src+16ws}$, $\mathbf{p}^{src+32ws}$, and $\mathbf{p}^{src+56ws}$, respectively.
> For instance, when $\mathbf{p}^{src}$ is "A living area with a television and a table", $\mathbf{p}^{src+8ws}$ would be extended to "A living area with a television and a table, highly detailed and precision with extreme detail description".
>
> We use Clipscore to evaluate that the generated images match the given prompt. In this case, we test our model under various length prompts ($\mathbf{p}^{src}$,$\mathbf{p}^{src+8ws}$, $\mathbf{p}^{src+16ws}$, $\mathbf{p}^{src+32ws}$, and $\mathbf{p}^{src+56ws}$) (see Tabel 1).
> As shown in Tabel 1, the generated images corresponding padding replacement prompts also contain similar semantic information compared to the initial prompt.
> The degeneration of the Clipscore is small (less than 0.11), indicating that the padding token also contains semantic information. Fig.21 (see in Appendix D. Fig.21) shows some more qualitative results.
>
> Table 1: Comparison results with original tokens and their replacement version. We evaluate it with Clipscore.
> | Method                    | $\mathbf{p}^{src}$ | $\mathbf{p}^{src+8ws}$ | $\mathbf{p}^{src+16ws}$ | $\mathbf{p}^{src+32ws}$ | $\mathbf{p}^{src+56ws}$ |
> |---------------------------|---------------------|-------------------------|--------------------------|--------------------------|---------------------------|
> | SD                        | 0.8208              | 0.8173                  | 0.8162                   | 0.8102                   | 0.8058                    |
> | SD w/ replacement         | 0.7674              | 0.7505                  | 0.7479                   | 0.7264                   | 0.7035                    |
>
>
>
> Related work also consider the padding embedding for other tasks. For example,  P2P  manipulates the padding  attention injection when conducing image-to-image translation.  P2P [1] swaps whole tokens attention,  including both the tokens and padding token attentions.
>
> **References**
>
> [1] Amir Hertz, Ron Mokady, Jay Tenenbaum, Kfir Aberman, Yael Pritch, and Daniel Cohen-Or. Prompt-to-prompt image editing with cross attention control. arXiv preprint arXiv:2208.01626, 2022

---

> ### Author Response · Authors · 2023-11-22
> **Response to Reviewer B9jb (2/3)**
>
> **1.2. Robustness of our method to various prompt lengths.**
>
> We use an object detection method to investigate the behavior of negative targets when using text prompts of various lengths.
> We first randomly select 50 real images and their corresponding prompts $\mathbf{p}^{src}$ from the datasets (a pool), as mentioned in Sec.4 of our main paper.
> The text prompts include various syntactical structures, such as "Girl holding toothbrush", "A woman wearing sunglasses and scarf" and "An airplane above a bridge".
>
> We use MMDetection with GLIP and the negative prompt to detect the probability of the negative target being present in both the input real image and edited image (DetScore). The DetScore are 0.4642 and 0.0393 (see Tabel 2 (second row, second and second column)), respectively.
> To investigate the behavior of negative prompts with longer sentences, we add description words with lengths 8, 16, and 32 following the initial text prompt $\mathbf{p}^{src}$, referred to as $\textbf{p}^{src+8ws}$, $\textbf{p}^{src+16ws}$, and $\textbf{p}^{src+32ws}$, respectively. For instance, when $\mathbf{p}^{src}$ is "Girl holding toothbrush", $\mathbf{p}^{src+8ws}$ would be extended to "Girl holding toothbrush, highly detailed and precision with extreme detail description".
>
> As shown in Table 2, although our method is a little more challenging to suppress the negative target in long sentences than in short ones, we can still efficiently suppress the negative target in long sentences.
> We also conducted the same experiments on generated images, and they showed the same pattern as in real images (see Table 3).
>
> Table 2: The average prediction score of MMDetection with GLIP using prompt negative prompt for real-image. Lower is better.
> | Method                   | SD     | Ours $\mathbf{p}^{src}$ | Ours $\mathbf{p}^{src+8ws}$ | Ours $\mathbf{p}^{src+16ws}$ | Ours $\mathbf{p}^{src+32ws}$ |
> |--------------------------|--------|--------------------------|-----------------------------|------------------------------|------------------------------|
> | DetScore$\downarrow$     | 0.4642 | **0.0393**               | 0.1746                      | 0.1649                       | 0.1674                       |
>
>
> Table 3: The average prediction score of MMDetection with GLIP using prompt negative prompt for generated-image. Lower is better.
> | Method                   | SD     | Ours $\mathbf{p}^{src}$ | Ours $\mathbf{p}^{src+8ws}$ | Ours $\mathbf{p}^{src+16ws}$ | Ours $\mathbf{p}^{src+32ws}$ |
> |--------------------------|--------|--------------------------|-----------------------------|------------------------------|------------------------------|
> | DetScore$\downarrow$     | 0.5619 | **0.1220**               | 0.1275                      | 0.1876                       | 0.1741                       |

---

> ### Author Response · Authors · 2023-11-22
> **Response to Reviewer B9jb (3/3)**
>
> **2. Taking a simple mean of the padding embeddings**
>
> We agree that all padding tokens are derived from the same input word embedding (e.g., the word embedding for text index 49407 in SD), with only positional embedding added to the word embedding. Subsequently, these embeddings are fed into the attention and MLP layers in CLIPTextEncoder. Ultimately, the padding token contains information about the entire prompt.
>
> In this paper, we attempt to extract the semantic component corresponding to the negative prompt target from the padding embeddings using Eq.2 in our main paper.
> Based on our observation (Sec. 3), the \textit{top-K} singular values in the constructed negative target embedding matrix $\boldsymbol\chi = [\boldsymbol{c}^{NE},\boldsymbol{c}^{EOT}\_0, \cdots, \boldsymbol{c}^{EOT}\_{N-{|\boldsymbol{p}|-2}}]$ mainly resides the content in the expected suppressed embedding $\boldsymbol{c}^{NE}$. Therefore, we utilize the formula ${e}^{-\sigma}*\sigma$ to ensure that the components corresponding to larger singular values undergo more shrinkage.
>
> We also perform the experiment abased reviewer's comment: using the mean of the padding embedded. Specially, we extract the semantic component by taking a simple Mean of the Padding Embeddings, referred as $\textbf{MPE}$.
> We evaluate the propsed  method (i.e., SVD) and  $\textbf{MPE}$. We suppress "glasses" subject from 1000 randomly generated images with the prompt "A man without glasses". Then we use MMDetection detect the probability of glasses  in the generated images. Final, we report the prediction score (DetScore).
>
> As reported in Table 4 (the third and fourth columns),  we have  $\textbf{0.1065}$  MMDetection score, while MPE is 0.6266.  This finding suggests that simply averaging the padding embeddings often fails to extract the main semantic component.
> Furthermore, we further zero the 'glasses'  token embedding as well as  MPE, it still struggles to extract 'glasses' information (0.4892  MMDetection).
> Fig22.(see in Appendix D. Fig.22) qualitatively shows more results.
>
> Table 4: Comparison between ours and MPE. We report Clipscore.
> | Method                            | SD      | Ours    | MPE     | MPE + zeroing embedding |
> |-----------------------------------|---------|---------|---------|--------------------------|
> | DetScore $\downarrow$             | 0.8052  | **0.1065** | 0.6266  | 0.4892                   |
>
>
>
> **3. Even if the attention map values are largely suppressed, the undesired semantics may still slip into the image features through the value recombination**
>
> We propose inference-time embedding optimization to further suppress the negative target generation and encourage the positive target content, following soft-weighted regularization.
> This optimization method involves updating the whole text embedding, which is then transferred to both the key and value components in the cross-attention layer.
> Therefore, our method implicitly changes the value component in the cross-attention layer.
>
> Furthermore, similar to the proposed two attention losses in the main paper, we attempt to use two value losses to regulate the value component in the cross-attention layer:
>
> $\mathcal{L}\_{vl} = \lambda\_{pl}\mathcal{L}\_{pl} +\lambda\_{nl}\mathcal{L}\_{nl}, $
>
> $\mathcal{L}\_{pl} = \left \|\boldsymbol{\hat{V}^{PE}\_t}-\boldsymbol{V^{PE}\_t}\right \|^2,$
>
> $\mathcal{L}\_{nl}= -\left \|\boldsymbol{\hat{V}^{NE\}_t}-\boldsymbol{V^{NE}\_t}\right \|^2,$
>
> where hyper-parameters $\lambda_{pl}$ and $\lambda_{nl}$ are used to balance the effects of preservation and suppression of the value. When utilizing this  value loss,
> we find that it is hard to generate high-quality images images (Fig.23 in Appendix D (the third and sixth columns)). This result indicates that directly optimizing the value embedding does not work. The potential reason is that it also influences positive target,  since each token embedding  contains other token embedding information after CliptextEncoder.
>
> **4. The CLIPscores in Table 1 are a bit confusing, do they represent the similarity between the images and negative prompts?**
>
> In our paper, Clipscore is a metric that evaluates the quality of a pair of a negative prompt
> and an edited image. We have updated it in our updated paper.

---

> ### Author Response · Authors · 2023-11-23
> **Response to Reviewer B9jb**
>
> As the rebuttal window is closing soon, we genuinely appreciate your feedback. We hope our response has addressed your concerns, and we kindly ask for your consideration in improving the rating of our paper. Thank you again for your time and contribution.

---

> > ### Comment · Reviewer_B9jb · 2023-11-23
> > **Thanks for the extra experiments**
> >
> > These experiments addressed most of my concerns. Although I still feel the conclusions are a bit counter-intuitive, I'd like to raise my rating to 6, before seeing more cross validation from the community.

---

> > > ### Author Response · Authors · 2023-11-23
> > >
> > > Thanks for your valuable suggestions and positive feedback regarding our work!

---

### Official Review · Reviewer_XjcJ · 2023-11-01

**Soundness:** 3 good
**Presentation:** 3 good
**Contribution:** 3 good
**Rating:** 6
**Confidence:** 3

**Summary:**

Aiming at supressing specific subjects from generated or real images, this work explores the hidden information of [EOT] embeddings in depth. Base on the discoveries, the soft-weighted regularization and inference-time text embedding optimization are proposed, enabling image editing without training or fine-tuning any large diffusion model. This light-weighted methology is effective and can be adapted to many applications, thus I believe it gives a solid and valuable contribution. The organization and phrasing of the article is also clear and easy to understand.

**Strengths:**

This light-weighted methology is effective and can be adapted to many applications, thus I believe it gives a solid and valuable contribution. The organization and phrasing of the article is also clear and easy to understand.

**Weaknesses:**

The diffusion model is a hot topic in machine learning and computer vision community. The differences should be further highlighted.

**Questions:**

With this methodology, we can remove subjects from an image or add subjects to it. Is it possible to change one subject to another in one go? For example, can we change the “toothbrush” in “Girl holding toothbrush” image to a “pen”?

**Details Of Ethics Concerns:**

There are no any concerns.

---

> ### Author Response · Authors · 2023-11-22
> **Response to Reviewer XjcJ**
>
> Thanks for your constructive comments.
>
> **1. The diffusion model is a hot topic in the machine learning and computer vision community, and further highlighting its differences is essential**
>
> Indeed, diffusion models are a hot topic, and one of the most impactful applications is their usage for text-guided image editing. The ability to create realistic images that faithfully follow user-provided prompts is of great importance for many applications. However, existing latent-space diffusion models struggle to effectively suppress the generation of undesired content, which is explicitly requested to be omitted from the generated image in the prompt. Therefore, in our method, we aim to address this problem and propose an improved method for negative target suppression.
>
> **2. Is it possible to change one subject to another in one go?**
>
> Subject replacement is a common task in various image editing methods [1-4].
> We can edit an image by replacing subject with another using only the prompt (see in Appendix F. Fig.31 (the second, fourth, and sixth columns)).
> We replace the text of the edited subject in the source prompt with the desired one to create the target prompt. Subsequently, we translate the real image using the target prompt into latent code. We then apply the same strategy as in the subsection on Appendix.F (**Attend-and-Excite similar results**) to obtain the edited image.
> For example, we can replace the "toothbrush" in the "Girl holding toothbrush" image with the "pen".
> The DetScore with "toothbrush" of the source image is 0.79, and the Clipscore with "pen" of the target image is 0.728.
> We find that the brace is also replaced with the pen, as the cross attention map for "toothbrush" couples the toothbrush and the brace.
> We will add subject replacement results to our paper in the final version.
>
> **References**
>
> [1] Chenlin Meng, Yang Song, Jiaming Song, Jiajun Wu, Jun-Yan Zhu, and Stefano Ermon. Sdedit: Image synthesis and editing with stochastic differential equations. arXiv preprint arXiv:2108.01073, 2021.
>
> [2] Gaurav Parmar, Krishna Kumar Singh, Richard Zhang, Yijun Li, Jingwan Lu, and Jun-Yan Zhu. Zero-shot image-to-image translation. arXiv preprint arXiv:2302.03027, 2023.
>
> [3] Ron Mokady, Amir Hertz, Kfir Aberman, Yael Pritch, and Daniel Cohen-Or. Null-text inversion for editing real images using guided diffusion models. arXiv preprint arXiv:2211.09794, 2022.
>
> [4] Narek Tumanyan, Michal Geyer, Shai Bagon, and Tali Dekel. Plug-and-play diffusion features for text-driven image-to-image translation. In Proceedings of the IEEE/CVF Conference on Computer Vision and Pattern Recognition, pp. 1921–1930, 2023

---

> > ### Comment · Reviewer_XjcJ · 2023-11-23
> >
> > Most of my concerns have been addressed and I keep my score 6.

---

> > > ### Author Response · Authors · 2023-11-23
> > > **Response to Reviewer XjcJ**
> > >
> > > Thanks for your valuable suggestions and positive feedback on our paper and rebuttal.

---

### Official Review · Reviewer_69SC · 2023-11-02

**Soundness:** 2 fair
**Presentation:** 4 excellent
**Contribution:** 3 good
**Rating:** 6
**Confidence:** 5

**Summary:**

This paper focuses on a sub-question of image editing, suppress the generation of undesired content, and proposes improvements from two aspects. Firstly, after conducting an analysis on text embeddings post text encoder encoding, it is concluded that EOT is low-rank and contains a large volume of prior information. Therefore, it also includes the information desired to be suppressed during the image editing process. Consequently, the first improvement proposed is to suppress the negative information in the text embedding to restrain its expression during the generation process. The second enhancement optimizes the attention map during the inference process to be as close as possible to the information to be preserved, while distancing from the unwanted information. Judging by the results, optimal outcomes were achieved on most datasets.

**Strengths:**

1. The analysis of the information components within text embedding provides certain guidance for subsequent T2Anything related research.
2. Judging by the results presented in the paper, it has achieved a rather precise suppression of information from the text, also outperforming previous works on numerical indicators.
3. There is no need for additional data training; any existing T2I model can be utilized.
4. The logic of the work is clear, and the exploratory part of the experiment is plentiful.

**Weaknesses:**

1. From the algorithmic perspective, both improvement points are existing methods, and thus lack a certain level of novelty.
2. This method requires gradient back-propagation during the inference process. Considering memory and time consumption, it doesn't seem as efficient as truly training-free methods like P2P.

**Questions:**

1. In the comparative experiments, it would be beneficial to specifically list the time and memory consumption ratios of this method compared to other methods, as this is necessary for a more application-oriented task.
2. In the first phase, this method uses coefficients to adjust the size of the negative information matrix to suppress the expression of negative information. If the singular value decomposition method is not employed, but instead, the entire matrix is multiplied by an attenuation factor, how would that affect the image editing results?
3. From Table 3, it seems that 'the negative target prompt suppression loss' plays the most significant role. What would be the effect if only this loss is considered without incorporating any other improvements?

---

> ### Author Response · Authors · 2023-11-22
> **Response to Reviewer 69SC (1/2)**
>
> Thanks for your constructive comments.
>
> $\textbf{1. From the algorithmic perspective, both improvement points are existing methods and thus lack a certain level of novelty}$
>
> We would like to stress that for efficient text-guided image generation, the usage of negative lexemes is very important. They are known to be essential for humans to precisely communicate the desired image content. Addressing this problem for text-guided image generation is therefore of great importance to the community and the wide application of these methods.
>
> Recently, both the pixel-space diffusion models (i.e.
> DeepFloyd-IF) and the latent-space diffusion models struggle to effectively suppress the generation of undesired content, which is explicitly requested to be omitted
> from the generated image in the prompt.  To achieve this goal, our work makes the following contributions: (I) Our analysis shows that the [EOT] embeddings contain significant, redundant and duplicated semantic information of the whole input
> prompt (the whole embeddings). This needs to be taken into account when removing negative target
> information. Therefore, we propose soft-weighted regularization to eliminate the negative target
> information from the [EOT] embeddings.  Although SVD is a well-known approach, to our best knowledge, we are the first to use it to address this suppression problem. (II) To further suppress the negative target generation,
> and encourage positive target content, we propose inference-time text embedding optimization. While attention mechanisms in our approach are known to the community for other tasks, it requires careful algorithmic design and a significant effort to incorporate the ideas of text embedding suppression into a single generative architecture.
>
> **2. This method requires gradient back-propagation during the inference process. Considering memory and time consumption, it doesn't seem as efficient as truly training-free methods like P2P**
>
> Note that, we aim to suppress the negative target generation in diffusion models (which is not addressed by e.g. P2P). To better compare the computational demands and memory requirements, we have added a table. We randomly select 100 prompts and feed them into the SD model.  As shown in the following table, we report the average values for inference time (s/image) and GPU memory demand (GB), respectively. Although we need additional time and computational overhead compared to P2P, we have better results in FID, Clipscore and DetScore metrics (see Table 1 in our main paper). For example, P2P has a 0.3391   DetScore, while our method is 0.1321.  While P2P does not require optimization during the inference stage and consumes less memory and time compared to ours, this method results in inferior performance (see Fig.6 in main paper).
>
> $\textbf{3. The time and memory consumption ratios of this method compared to other methods}$
>
> In the following table, we report  the time and memory consumption ratios.   We randomly select 100 prompts and feed them into the SD model.  Compared to the baselines, we need additional time and memory consumption.  Note that, we aim to suppress the negative target generation in diffusion models. In  the Table 1 in our main paper,   For real-image negative target suppression,  we achieve the best score in both Clipscore and DetScore.
>
> Table 1: Computational cost evaluation in baselines and ours.
> | Method            | SD     | Negative Prompt | P2P     | ESD     | Concept-ablation | Forget-Me-Not | Inst-Inpaint | SEGA    | Ours    |
> |-------------------|--------|-----------------|---------|---------|-------------------|---------------|--------------|---------|---------|
> | Inference time (s/image)    | 7.35   | 7.42            | 15.434  | 14.407  | 5.240             | 13.231        | 1.093        | 20.157  | 24.617  |
> | GPU Memory demand (GB)  | 12.3   | 12.3            | 12.4    | 9.9     | 9.7               | 21.8          | 9.8          | 11.8    | 14.3    |
> | Tuning-free       | ✔      | ✔               | ✔       | ✘      | ✘                 | ✔             | ✘           | ✔       | ✔       |
> | Image resolution  | 512x512| 512x512         | 512x512 | 512x512 | 512x512           | 512x512       | 256x256      | 512x512 | 512x512 |

---

> > ### Author Response · Authors · 2023-11-22
> > **Response to Reviewer 69SC (2/2)**
> >
> > **4. If the singular value decomposition method is not employed but instead the entire matrix is multiplied by an attenuation factor, how would that affect the image editing results?**
> >
> > We evaluate the advised method involving an attenuation factor. We experimentally observed that employing an attenuation factor (e.g., 0.1) for the negative target embedding matrix would impact the positive target (see in Appendix D. Fig.20). Hence, using an attenuation factor leads to unexpected subject changes as well as changes to the target subject. This is due to the fact that the [EOT] embeddings contain significant information about the input prompt, including both the negative target and the positive target (see Sec 3.2). Furthermore, the selection of factors needs to be carefully performed for each image to achieve satisfactory suppression results.
> >
> > **5. From Table 3, it seems that 'the negative target prompt suppression loss' plays the most significant role. What would be the effect if only this loss is considered without incorporating any other improvements?**
> >
> > As presented in $\textbf{Subsecton 3.4 INFERENCE-TIME TEXT EMBEDDING OPTIMIZATION}$, to compute  $\mathcal{L}\_{nl}$, we need first to perform SWR to generate both the original text embedding $\boldsymbol{c}$ and the modified text embedding $\boldsymbol{\hat{c}}$. Thus we have to combine SWR to evaluate  $\mathcal{L}\_{nl}$.  As shown in Fig 13. in the Appendix D, the regions that are not expected to be suppressed are structurally altered with SWR + $\mathcal{L}\_{nl}$ (third row).  When using $\mathcal{L}\_{nl}$ and $\mathcal{L}\_{pl}$,  our method removes the subject while mainly preserving the rest of the regions (fourth row)

---

### Official Review · Reviewer_9uRw · 2023-11-05

**Soundness:** 3 good
**Presentation:** 3 good
**Contribution:** 3 good
**Rating:** 6
**Confidence:** 4

**Summary:**

This paper addresses the challenge of controlling the generation of unwanted content in text-to-image diffusion models by introducing two methods: soft-weighted regularization and inference-time text embedding optimization. These techniques effectively suppress undesired content and encourage the generation of desired content, with positive results demonstrated through quantitative and qualitative experiments on both pixel-space and latent-space diffusion models.

**Strengths:**

1. The introduction of "soft-weighted regularization" that effectively removes negative target information from text embeddings, improving the control over undesired content generation.

2. The method is more efficient than previous methods: no need for fine-tuning the generator and no collection of paired images.

3. An interesting and inspiring analysis is conducted in Section 3.2.

**Weaknesses:**

1. This work introduces some new matrix computations, such as the SVD in soft-weighted regularization and the attention map alignment in ITO. However, the authors do not discuss the additional computational overhead of these computations.

2. In Tab. 1, the proposed method is outperformed by baselines on some metrics under certain settings. It would be better to analyze why this occurs.

**Questions:**

Please address questions in "Weaknesses".

---

> ### Author Response · Authors · 2023-11-22
> **Response to Reviewer 9uRw**
>
> Thanks for your constructive comments.
>
> $\textbf{1. The additional computational cost of SWR and ITO}$
>
> We randomly select 100 prompts and feed them into the SD model.  As shown in the following Table, we report the average values for inference time (s/image) and GPU memory demand (GB), respectively. For a comparison of time and memory consumption ratios with other methods, also see the third question ( REVIEWER2 69SC).
>
>
> Table: Computational cost evaluation for Inference time (s/image).
> | Method | SWR | ITO | GPU Memory demand (GB) |
> |----------|----------|----------|----------|
> | Ours  | 0.134  | 24.483  | 14.3  |
>
>
> $\textbf{2. In Tab. 1, the proposed method is outperformed by baselines on some metrics under certain settings}$
>
> For the real-image editing setting, although $\textit{negative prompt}$ has the best IFID score (175.8), our method has a very similar IFID score (166.3).  However, $\textit{negative prompt}$ has the worst 0.2402 DetScore, and a very wide distance with our method (0.0384 DetScore).  Furthermore,   $\textit{negative prompt}$  often changes the structure
> and style of the image (Fig. 6 (left, second row)). In contrast, our method achieves a better
> balance between preservation and suppression (Fig. 6 (left, last row)).
>
> For the generated-image editing setting for $\textit{Tyler Edlin}$, ESD leads to the best performance, such as (Clipscore, IFID): (0.6954, 256.5).  ESD utilizes negative guidance to lead the fine-tuning of a pre-trained SD model. However, it results in
> catastrophic neglect of the negative target prompt [1]. Thus, ESD cannot use the negative prompt for future editing tasks. In this paper, we aim to remove unwanted subjects from output images without further training or fine-tuning the SD model. Please see the detailed presentation in Sec. 4. experiments.
>
> $\textbf{References}$
>
> [1] Nupur Kumari, Bingliang Zhang, Richard Zhang, Eli Shechtman, and Jun-Yan Zhu. Multi-concept customization of text-to-image diffusion. arXiv preprint arXiv:2212.04488, 2022.

---

### Meta-Review · Area_Chair_AxGJ · 2023-12-05

**Metareview:**

The paper presents two methods for reducing unwanted content in text-to-image models: soft-weighted regularization and text embedding optimization. It focuses on managing negative information in text embeddings and refining attention maps to enhance desired content. Additionally, it employs Singular Value Decomposition to address side effects from padding tokens and introduces specific losses for target management. These lightweight approaches have proven effective in various tests, improving content control without extensive model training.

**Justification For Why Not Higher Score:**

The reviewers have expressed concerns about the technical novelty of this paper, noting that the major improvements it introduces are largely derived from existing literature. However, the authors have responded, clarifying that while attention mechanisms are familiar to the community in other contexts, integrating text embedding suppression into a single generative architecture involves meticulous algorithmic design and substantial effort. Additionally, regarding the effects of padding tokens, the reviewers found the conclusions somewhat counter-intuitive. In response, the authors have conducted more systematic experiments using prompts of varying lengths to provide clearer insights.

**Justification For Why Not Lower Score:**

The paper contributes to text-to-image research by analyzing text embeddings, offering insights for future studies. It shows competence in suppressing unwanted text information and performs better than previous methods in certain numerical aspects. A key advantage is its compatibility with existing T2I models, eliminating the need for extra data training. The examination of semantics in padding tokens is a noteworthy aspect. Additionally, the designed losses for positive preservation and negative suppression are straightforward and contribute to the paper's practical value.

---

### Decision · Program_Chairs · 2024-01-16

Accept (poster)